# The key role of sufficiency for low demand-based carbon neutrality and energy security across Europe

Frauke Wiese [1,15] ✉, Nicolas Taillard [2,15], Emile Balembois [2], Benjamin Best[3], Stephane Bourgeois [2], José Campos [4], Luisa Cordroch [1], Mathilde Djelali[2], Alexandre Gabert[2], Adrien Jacob [2], Elliott Johnson[5], Sébastien Meyer [6,7], Béla Munkácsy [4], Lorenzo Pagliano [8], Sylvain Quoilin[9], Andrea Roscetti [8,10], Johannes Thema [1,11], Paolo Thiran [7], Adrien Toledano[2], Bendix Vogel[1,12], Carina Zell-Ziegler [13,14] & Yves Marignac [2]

A detailed assessment of a low energy demand, 1.5 °C compatible pathway is provided for Europe from a bottom-up, country scale modelling perspective. The level of detail enables a clear representation of the potential of sufficiency measures. Results show that by 2050, 50% final energy demand reduction compared to 2019 is possible in Europe, with at least 40% of it attributable to various sufficiency measures across all sectors. This reduction enables a 77% renewable energy share in 2040 and 100% in 2050, with very limited need for imports from outside of Europe and no carbon sequestration technologies. Sufficiency enables increased fairness between countries through the convergence towards a more equitable share of energy service levels. Here we show, that without sufficiency measures, Europe misses the opportunity to transform energy demand leaving considerable pressure on supply side changes combined with unproven carbon removal technologies.

Most energy scenarios focus on technical strategies to achieve decarbonisation at the lowest cost and pay little attention to the environmental and social consequences of these strategies other than the reduction of greenhouse gas (GHG) emissions[1–3]. Consequently, important dynamics of a sustainable transformation are ignored, such as the social and ecological consequences of resource extraction and energy imports that are necessary for the respective scenario. With an increasingly broader focus on other sustainability dimensions beyond climate protection, demand-side climate mitigation solutions are receiving increased attention[4] as they increase the probability of reaching climate ambitions and have high synergies with other Sustainable Development Goals[5]. However, most techno-economic optimisation approaches ignore demand-side and, in particular, sufficiency strategies. The energy sufficiency concept[6–8] refers to reducing, in absolute terms where adequate, the consumption and production of end-use products and services through changes in social

[1]Department of Sustainable Energy Transition, Europa-Universität Flensburg, Flensburg, Germany. [2]négaWatt Association, BP 16280 Alixan, 26958 VALENCE Cedex 9, France. [3]City of Bonn, Climate Neutral Bonn 2035 Program Office, Bonn, Germany. [4]Department of Environmental and Landscape Geography, ELTE University, Budapest, Hungary. [5]Sustainability Research Institute, University of Leeds, Leeds, UK. [6]negaWatt Belgium, Rue du Blanc-Ry 163, Ottignies-Louvain-la-Neuve 1342, Belgium. [7]Institute of Mechanics, Materials and Civil engineering, Université Catholique de Louvain, Louvain-la-Neuve, Belgium. [8]Architecture and Urban Studies Department, Politecnico di Milano, Italy. [9]Integrated and Sustainable Energy Systems, University of Liege, Liege, Belgium. [10]Università della Svizzera italiana, Accademia di Architettura, Architettura, Switzerland. [11]Energy, Transport and Climate Policy Division, Wuppertal Institute for Climate, Environment and Energy, Wuppertal, Germany. [12]Potsdam Institute for Climate Impact Research (PIK) e.V., Potsdam, Germany. [13]Department of Landscape Planning and Development, Technische Universität Berlin, Berlin, Germany. [14]Energy & Climate Division, Oeko-Institut, Berlin, Germany. [15]These authors contributed equally: Frauke Wiese, Nicolas Taillard. ✉e-mail: frauke.wiese@uni-flensburg.de

practices (supported by adequate infrastructures and frameworks), to remain within planetary boundaries while ensuring a decent social foundation for all people. This social foundation commonly refers to the idea of decent living standards[9]. Millward-Hopkins et al.[10] have suggested minimum energy service levels per person that comply with decent living standards and would result in 60% lower global final energy consumption in 2050, despite the population being 30% larger. This indicates a lower boundary for the sufficiency corridor. On the other side, Grubler et al.[11] estimate a 40% reduction in total global final energy in 2050 compared to 2020 worldwide that is necessary to comply with a 1.5 °C climate target without negative emission technologies. The reduction in their study is partly caused by increases in energy conversion efficiency and partly by 'use efficiency', which is a combination of organisational, institutional and infrastructural factors. Sufficiency levers were included to a lesser degree, e.g., a reduction in passenger-km/capita in the Global North and increases in the Global South. Barrett et al.[12] investigated the potential of energy demand reduction in a case study for the United Kingdom. Using a bottom-up approach, the energy scenario demonstrates potential energy demand reductions of 52% by 2050 in comparison to 2020, via a combination of efficiency and sufficiency measures. This is achieved while increasing quality of life, thanks to the associated co-benefits in areas such as health, improved local environments, better work practices, reduced investment needs.

In this paper, we present a scenario at the European level with a strong focus on sufficiency options and exchange and transmission of hydrogen and power between European countries as an essential characteristic. This enables a strong reduction of dependency on energy imports from outside Europe as well as excluding the further use of nuclear and the introduction of carbon capture and storage technologies (CCS) by 2050. The so-called Collaborative Low Energy Vision for the European Region (CLEVER) scenario describes overall reductions in final energy consumption at the European level (EU30: EU27, plus UK, Norway and Switzerland) but suggests an increase for some energy service demands in Member States where per capita consumption is below the level considered sufficient (concept of convergence). Social and environmental components of sustainability are both considered as integral to this scenario. 50% of final energy demand reduction is realised (compared to 2019). At least 40% of this final energy demand reduction is due to sufficiency, offering short-term reductions that are essential to address cumulative emissions. Sufficiency is also essential to achieve deep sustainability and limit other socio-environmental impacts, such as the impacts of mining and the demand for materials, including (i) materials requiring energy-intensive industrial processes and (ii) other biotic or abiotic materials for non-energy uses. The methodology, a bottom-up collaborative modelling approach, is tailored to the national contexts but also coherent at the continental level and fills the gap in sufficiency scenarios between global and national level.

Key components of the theoretical framework behind the CLEVER scenario are: (1) rapid reductions in energy demand to address cumulative emissions (2) a fair distribution of the remaining carbon budget (3) improved energy, materials security and resilience (4) contribution to other sustainability principles beyond climate change mitigation and (5) enhanced political and economic collaboration in Europe. To comply with (1) the scenario is a 1.5 °C compatible pathway, which implies not only carbon neutrality in the 30 investigated countries of Europe, but also near-term, rapid GHG reductions to stay within the cumulative budget. Taking a 50% probability and per capita distribution of the global budgets (IPCC Sixth Assessment Report[13] AR6 I, SPM, p.29, Table SPM.2) results in 30–32 GtCO$_2$ as the maximal EU30 CO$_2$ budget for 2020–2050, which corresponds to 6% of the global budget (further information on the budget calculation can be found at Bourgeois et al.[14], p.18). The range refers to the various points of

cumulative emissions in 2045 and 2050, with cumulative emissions peaking at 32 GtCO$_2$ in 2045, before declining to 30 GtCO$_2$ in 2050 as a result of net negative emissions. A per capita distribution of the budget results in a fairer distribution of the remaining carbon budget, emphasising equity within the CLEVER scenario (2) and is consistent with the concept of common but differentiated responsibility as defined in the Paris Agreement. Improved energy and materials security (3) implies a quick reduction of sensitive energy and materials imports and the avoidance of additional dependencies in the form of energy carriers and materials. The base supply in the scenario is built on local resources, supporting exchanges within Europe (EU30). Nuclear power production, as a high-risk supply option contributing to import dependencies, is phased-out by 2050. To further comply with the key aspect of resilience, the pathway construction ensures a balance of ambition and realism regarding the pace of expansion of technology and infrastructure as well as depth of changes. Technologies with a technology readiness level (TRL) of at least 7 were privileged as those less developed currently are unlikely to be deployed at scale before 2050 or only at limited capacities. The choice of available GHG reduction options is also premised on a principle of deep sustainability (4) to deliver a scenario compatible with the other planetary boundaries beyond climate change mitigation[15]. A detailed consideration of the implications of these choices for other planetary boundaries is provided in the Supplementary Discussion. The key component of enhancing solidarity across Europe (5) results in a fair distribution of resources and contributions to mitigation by approaching a target level corridor. This leads to a convergence of energy service demands within the examined European states. In contrast to other climate-neutrality scenarios for Europe[16–20], energy demand reduction strategies beyond efficiency are considered to achieve multiple sustainability objectives.

The modelling approach involves several steps, beginning with the development of bottom-up national trajectories, followed by a comparison of the ambition levels of the national trajectories by one another and by other references to provide a basis for a harmonised European scenario and to check whether the ambitions are sufficient and realistic, harmonisation of the data and convergence of sufficiency indicators until 2050, and ultimately synthesis and contraction in an iterative process aimed at achieving European sustainability targets. National trajectories have been developed by the active partners (for an overview of organisations, countries and roles see Bourgeois et al.[14], p.6,13), either from modelling specifically dedicated to the scenario or from existing scenarios, and detailed national data was collected in a common dashboard template. For other European countries, normalised (and conservative) trajectories were developed and reviewed by national experts. Using data from the literature, target level corridors for key indicators such as heated living space per capita or passenger-kilometres per capita were defined, with a minimum floor, among others, shaped by decent living standards[9,10] and the maximum on 1.5 °C compatible service levels[11]. Decent living standards define minimum living standards on an individual level (i.e., activity levels per capita). At the national level, total energy consumption projections based on these activity levels would reflect a theoretical minimum level of consumption for a given country. However, due to intra-country inequality of service-level-indicators (e.g., m²/cap), this would result in a proportion of the population falling below decent living minimum standards. We thus do not apply theoretical minima, but set national average activity-levels to exceed decent living thresholds to account for within-country inequalities. Sufficiency indicators touch upon dimensional (e.g., size of vehicles), service-related (intensity and duration of use of vehicles) or organisational (e.g., as the development of collective transport) aspects of energy consumption. Exceptions to the target corridor levels were made in cases of justified national specificities. Table 1 provides a summary of main sufficiency indicators applied in CLEVER.

**Table 1 | Overview of main energy sufficiency indicators applied in the CLEVER scenario**

| Sector | Indicator | Unit | Explanation |
|---|---|---|---|
| Mobility | Passenger transport demand | pkm/cap per year | Distance travelled per capita: Number of kilometres travelled per person and per year |
| | Passenger transport demand: Plane | pkm/cap per year | Number of kilometres travelled by air per person and per year (domestic and international) |
| | Share of active mobility | pkm/cap per year or% | Distance travelled per person and per year via active modes (mainly cycling and walking) expressed in passenger-kilometres or as a share of total distance travelled per capita |
| | Share of collective transports | % | Share of domestic distance travelled per capita (pkm), except air and active mobility, travelled by bus, coach, metro/tram, train or boat |
| | Car occupancy | person/car | Average number of passengers per car travelling (weighted average over all segments of cars) |
| Freight | Domestic freight transport demand | tkm/cap per year | Domestic freight amount and distances: Tons times kilometres transported per year divided by the population |
| | Share of rail transport | tkm/cap per year or% | Part of domestic freight transport by rail |
| | International maritime freight transport | tkm/cap per year | International maritime freight amounts and distances: Tons times kilometres transported on international waterways to the country per year divided by the population |
| Residental | Living space | m²/person | Useful floor space of dwellings permanently occupied, divided by the population |
| | Domestic hot water FEC | kWh/person per year | Final energy consumption per person for hot water per person per year; mix of sufficiency (hot water needs per person) and efficiency of the water heating system |
| | FEC for specific electricity | kWh/person per year | Final energy consumption per person for specific electricity; mix of sufficiency (size and number of equipments; frequency/duration of use) and efficiency (energy performance of the equipment) |
| Industry | Production / demand of a material | Index (2015) | Evolution by comparison to 2015 of the production of several materials (cement, steel, pulp/paper, glass, ammonia and high value chemicals); no relocalisation/delocalisation assumptions have been made. The ratio between national demand and national production is supposed to remain stable. Then an evolution of demand (in%) is equals to the evolution of production |
| | FEC of an industrial sector | Index (2015) | Evolution by comparison to 2015 of the final energy consumption of an industrial sector (food, chemicals, non-ferrous metals and "Others"); mix of sufficiency (evolution of the demand, circularity) and efficiency of the process |
| Agriculture and Food | Meat consumption | g/day | Average daily meat consumption per person |
| | Consumption of dairy products | g/day | Average daily consumption of dairy products per person |

In this work, we present detailed national energy and emissions pathways for 30 European countries, consistent at both the country and regional level. Each country's pathway contains comprehensive information on energy service demands and carriers for each end use sector, as well as extensive energy production data, which can be used by both national- and European-level policymakers to guide policy and target-setting.

## Results

### Greenhouse gas emission reductions
By 2030 and 2040, net GHG emissions in the CLEVER scenario are cut by approximately 50% and 90%, respectively, compared to 2019 (3.9 GtCO₂eq.). Substantial GHG emission reductions (Fig. 1) in the scenario occur in the heat and power sector, residential sector and transport sector (which includes aviation). Remaining emissions (0.3 GtCO₂eq.) in 2050 are mostly from the industry and agriculture sector, and can be offset by natural carbon sinks through land and forestry management (approx. 0.5 GtCO₂eq.). Net zero emissions are reached by 2046 without a reliance on nuclear or dedicated CCS technologies, which aligns with other low energy demand scenarios[11,12,21,22]. These results are compatible with a European carbon budget for limiting global warming to 1.5 ℃ with a probability of 50%, based on an equal per capita approach distribution.

### Reduction in final energy and service demand
Strong reductions in final energy consumption are achieved through sufficiency and efficiency levers. Total final energy consumption is cut by 50.3% (final energy consumption for all energies, excluding ambient heat, maritime bunkers, the energy sector and non-energy uses[23]) from 2019 to 2050 (corresponding to a reduction from 13,250 TWh to

6590 TWh), with each country achieving per capita reductions of between 27% and 72% (Fig. 2), illustrating how Western European countries (i.e., those that currently have the highest energy consumption) are estimated to contribute the greatest towards demand reduction in the CLEVER scenario.

Figure 3 shows the results of the approach of combining strong reductions with fairness considerations for two key energy service demands in the residential buildings[24] and transport sectors[25]. The base year of 2019 is chosen to account for anomalous energy service levels in 2020 as a result of restrictions imposed by the global pandemic response.

### Sufficiency and efficiency impact in the industry sector
Sufficiency in the industrial sector consists of adjusting the nature and intensity of industrial demand to meet needs and services while minimising materials consumption. As industrial demand is created by other sectors (buildings, transport and food), the levers of sufficiency in these sectors (for example, reducing new construction leads to a reduction in industrial demand for steel and cement) are essential to achieving high levels of final energy demand reduction (Fig. 4). The production volumes are based on sufficiency-oriented consumption at the product and service level, but refer to other studies; they are not modelled at the product level in this study. The historic rate of production over consumption of material is kept in the CLEVER scenario.

### Supply side - high exchange within Europe, limited imports
The considerable reductions in final demand outlined above have significant implications for the transformation of energy supply and mitigate against many of the risks associated with the supply-side transition. Lower demand means that a much smaller energy system is

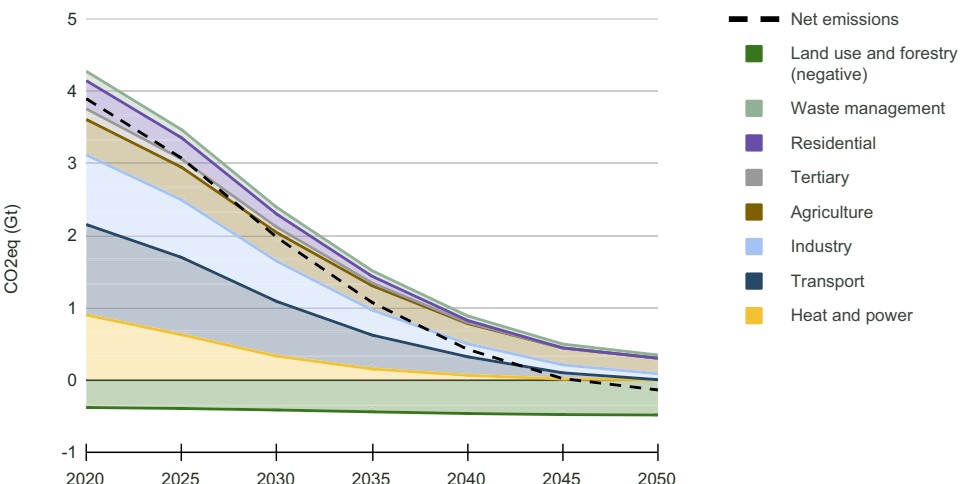

**Fig. 1 | Annual greenhouse gas emissions in the CLEVER (Collaborative Low Energy Vision for the European Region) scenario by emitting sector, 2019–2050 (EU30).** Carbon neutrality is reached before 2050. Remaining emissions are offset by natural sinks. The carbon budget is compatible with a European carbon budget for limiting global warming to 1.5 °C with a probability of 50%, based on an equal per capita approach distribution. Gt CO$_2$ eq: gigaton CO$_2$ equivalent. A list of countries included in EU30 can be found in Supplementary Table 7. Source data are provided as a Source Data file.

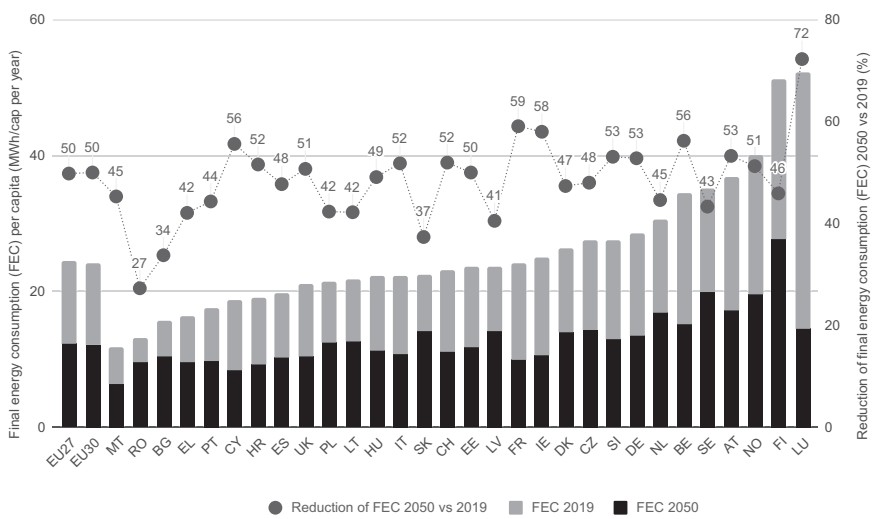

**Fig. 2 | Development of the final energy demand.** Annual final energy consumption (FEC) per capita (left axis) and percentage reductions in final energy consumption per capita (2019–2050, right axis) across the EU30 countries. For final energy consumption, we apply the definition of Eurostat[23] - excluding ambient heat, non-energy consumption, the energy sector and maritime bunkers from the total. A list of countries included in EU30 and EU27 and full names of countries can be found in Supplementary Table 7. Source data are provided as a Source Data file.

required, which allows for net zero emissions to be achieved with a much smaller-scale rollout of low-carbon technologies, limits the proliferation of the electricity grid and storage infrastructures, offers the possibility for Europe to almost completely remove its dependence on energy imports and limit dependence on imported materials. Demand can also be met whilst excluding high-risk supply-side technologies, such as nuclear power.

Figure 5 illustrates the difference in size and energy flows between the energy system for EU30 in 2019 and 2050. Renewable energy sources are scaled up from 3096 TWh/y in 2019, to 8837 TWh/y by 2050, with solar PV (1683 TWh/y, 1046% increase in capacity compared to 2019), onshore (1782 TWh/y, 384% increase) and offshore wind (1697 TWh, 2183% increase) making up the majority of production. Bioenergy contributes to supply, with solid biomass contributing towards 1296 TWh/y of primary demand (a 9% increase), biogas making up 619 TWh (213% increase) and liquid biomass contributing 205 TWh (7% increase). The sustainability issues relating to bioenergies

are documented in research, such as the scale of land use required, food security, water supply, biodiversity and social issues[26–28]. To minimise these problems, the scaling up of bioenergy would need to be realised without inducing unintended and undesirable land use changes[29]. Thus, a conservative approach was taken to bioenergies, with assumptions on bioenergy potentials on the low end of existing estimates for Europe - without relying on imports. For details on the bioenergy potential modelling, see Section Methods and Supplementary Method.

The remainder of primary demand in 2050 is supplied by miscellaneous renewables (marine, geothermal, hydro, solar thermal, ambient heat and waste heat recovery - 1539 TWh), petroleum (104 TWh) and waste (17 TWh non-renewable, 16 TWh renewable). Furthermore, total extra-EU imports are reduced tenfold, from 10.5 to 0.1 PWh from 2019 to 2050, with just 104 TWh of imported petroleum being required in 2050 as a feedstock for the olefins industry. The share of local primary energy production (i.e., energy produced within

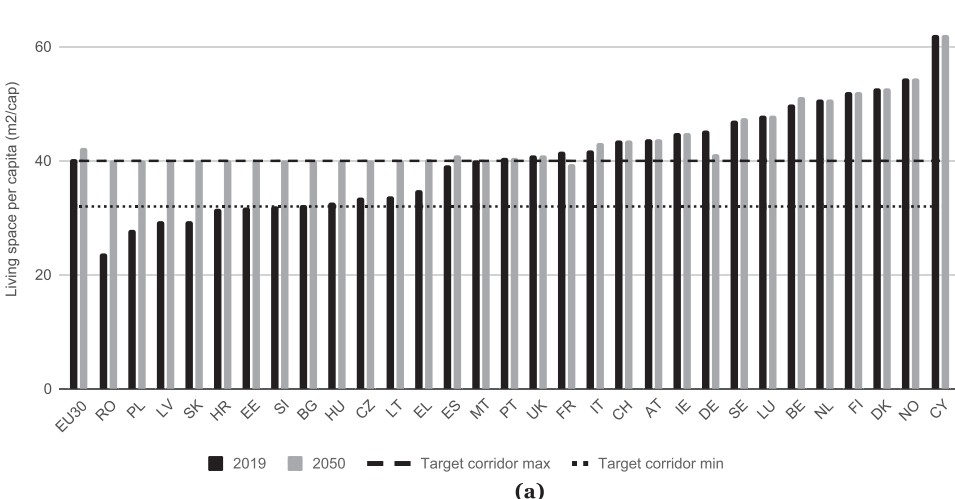

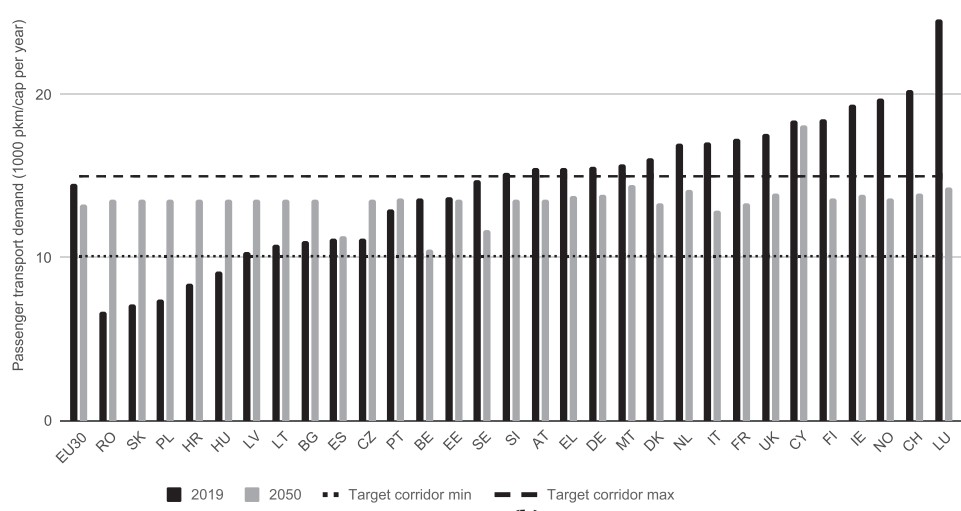

**Fig. 3 | Corridors and development of two energy service demand indicators.**
**a** Development of living space per capita in residential buildings in 2019 and 2050.
Units: square meter ((m²) per capita (cap) on average. An increase in residential
floor space [m²] per person is assumed for countries with smaller living spaces per
capita and capped at 40 m²/capita. However, due to the strong inertia inherent in
the building stock and trends that are difficult to reverse (larger houses and a
reduction in people per household), reversing the floor space per person was
deemed infeasible - it would imply a destruction of buildings without replacement.
Thus, the objective is to stabilise m²/cap for those countries that already exceeded
the upper limit of 40 m²/capita and prevent further increases in inequalities. Full
names of countries can be found in Supplementary Table 7. Source data are pro-
vided as a Source Data file. **b** Development of passenger transport demand. Units:
1000 passenger-kilometres (pkm) per capita (cap) and year on average. Here,

passenger traffic excludes soft modes (walking, cycling) but includes air traffic
(domestic and international). The convergence approach enables countries with
low levels of consumption below the minimum threshold (e.g., passenger traffic per
capita in Slovakia, Poland and Hungary, amongst others) to reach a level that is
consistent with decent living standards. High levels of energy service demands are
reduced through sufficiency levers, for example, passenger traffic in Italy decreases
from 18,513 pkm/cap in 2019 to 12,881 pkm/cap in 2050. Transgression of the upper
boundary of the consumption corridor is a result of international aviation in
countries with geographies that will require higher levels of demand for this mode
of transport (e.g., island countries) or have other national specificity. Full names of
countries can be found in Supplementary Table 7. Source data are provided as a
Source Data file.

Europe) increases greatly, from 44% in 2019 up to 99% in 2050. In the
CLEVER scenario, the basis of Europe's energy supply is affordable and
local production. Nuclear power production is slowly phased out by
2050, from 2297 TWh/y in 2015 without a corresponding increase in
coal demand.

As can be seen in Fig. 5, energy flows in total are reduced and
electricity is the backbone of the whole system. While hydrogen plays
an important role for industry and non-energy use, for the other

sectors it has a very limited role compared to electricity as an energy
carrier.

There are high levels of inter-European trading of hydrogen and
electricity (Fig. 6), but energy imports from outside of Europe are
almost entirely eliminated (Fig. 5). Crucially, EU30 countries are not
reliant on imports from outside of Europe for hydrogen or hydrogen
feedstocks, with Europe-wide production coverage reaching 107% of
demand, producing 755 TWh/y in 2050. By managing hydrogen

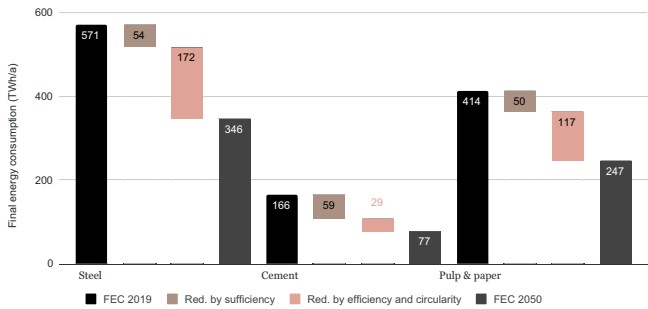

**Fig. 4 | Changes in final energy consumption (FEC) in three energy-intensive industrial subsectors in EU30.** Unit: Terawatt hours (TWh) per year (a). Demand reduction in cement industry is mainly driven by sufficiency (Red. by sufficiency), whereas demand reduction in steel and pulp & paper industries is driven by efficiency and circularity improvements (Red. by efficiency and circularity). In the vast majority of industrial sectors, circularity enables a reduction in unit energy consumption. When the applications of materials produced by recycling processes are the same as those from primary production, we could consider that circularity is part of efficiency measures if we restrict ourselves to the energy perimeter. However, a more systemic approach integrating the issue of reducing the consumption of mining products and so limiting the socio-environmental impacts of extractive activities and refining, could consider circularity as part of sufficiency. A list of countries included in EU30 can be found in Supplementary Table 7. Source data are provided as a Source Data file.

imports within Europe, there is no need for externalisation of generation needs to other world regions, which is a key element of a global just energy transition. Another aspect of a just energy transition that goes beyond the quantitative analysis of this paper, would be to consider the role of imported manufactured goods and not only the import of energy carriers as potential justice issues might continuously occur due to unequal exchange between European countries and the Global South.

## Discussion

The CLEVER scenario contributes to the growing research basis that is developing around the potential for energy demand reduction in developed countries. A comparison to other scenarios that prioritise reductions in energy demand, with varying geographical scopes (world: Low Energy Demand (LED) for the Global North[11], Transform scenario for the UK[12], 2050 energy scenario for France[22] and Germany's RESCUE GreenSupreme scenario[21]), demonstrates an ambitious range of potentials, coalescing around the −50 to −55% mark for final energy reduction. One consistent result across these scenarios is that a strong emphasis on energy demand reduction ensures that they can all reach net zero emissions without relying on CCS technologies.

Although the CLEVER scenario does not account for costs and investment requirements, previous research suggests that lower final demand achieved through sufficiency can limit the energy system costs of the transition to net zero emissions. Low energy demand results in a smaller energy system, meaning that less renewable supply-side and storage infrastructure is required to meet demand and displace fossil fuels[12,30]. Furthermore, lower demand results in fewer residual emissions, meaning that expensive technologies such as carbon capture are not required and do not inflate investment costs[12]. The CLEVER scenario also demonstrates increased integration of domestic European energy systems through high levels of hydrogen and power exchange via transmission, which has been shown to reduce costs in a 100% renewable European energy system[31]. Additionally, smaller energy system reduce materials requirements, embodied energy and emissions of the energy transition[22,32,33].

Sufficiency is adopted within the modelling framework at the international level but intra-country inequalities are not explored. By

reducing national average energy consumption, energy demand reduction has the potential to lead to the lowest consumers falling below decent living standards without addressing energy inequality[34]. The CLEVER scenario adopts sufficiency floors above that recommended limits, to allow for sufficient space for the lowest consumers. For example, assumptions for average residential floor space of approximately 42m²/capita (EU30) are between 2.1 and 2.8 times the recommended decent living minimum standards[10], allowing for a floor-space Gini of between 0.36 and 0.48, according to the framework developed by Pauliuk[35]. Similarly, for surface transport, the population-weighted average for surface travel in the CLEVER scenario for the countries assessed would permit a passenger-kilometres Gini coefficient of up to 0.38 in relation to decent living standards suggested by Millward-Hopkins et al.[10].

Future work should build upon this by assessing how variations in income inequality impact access to energy services and the resulting compatibility with decent living energy thresholds. Sufficient analysis also needs to be given to how policy measures must be designed not only to ensure that the sufficiency corridor is achieved on average per country, but also to ensure fair distribution within countries so that minimum standards are achieved for each person.

As for renewables and efficiency potentials, the realisation of sufficiency potentials and its fair distribution is contingent on a respective policy framework[4,36,37] partly in interdependence with deeper cultural and social changes. However, sufficiency does not play a major role in policy frameworks of existing scenarios yet, as renewable energies and efficiency are usually the dominant strategies in rather technology-focused narratives[38]. In the present scenario, policies were not modelled in detail, but instruments have been outlined for all sectors. In the Supplementary Data 1, about 100 policy measures are listed. Additionally, for a better overview, a summary of main policy strategies per sector pursued by sufficiency policies within the CLEVER scenario, and examples for specific instruments. Some of the policy measures are not only aimed at achieving the sufficiency corridor per country in Europe, but also explicitly at ensuring that every individual within the country achieves the basis for a good life.

Sufficiency policy mixes require a range of instrument types, regulative but also fiscal and information policies[39–41]. Effective policy mixes are characterised by consistency (to exclude contradictions), coherence (to use synergies between instruments) and comprehensiveness (to support the full range of strategies for a shift to a 100% renewable energy system)[42].

The instrument mix proposed to achieve the CLEVER scenario varies by sector and mostly targets structural changes (by regulation, economic instruments, education) rather than individual behaviour (information). In contrast to the present European National Energy and Climate Plans[39], all sectors include a large share of regulatory sufficiency measures (20%-50%). For an overview graph of instrument types per sector, see the Supplementary Fig. 5. This approach of mixed policy instruments resonates with the recommendations of Climate Citizen Assemblies in European countries, which additionally call for a regulatory framework in climate policy, where the state sets clear rules rather than relying on price and market instruments[40]. The strong focus of citizens' assemblies on sufficiency measures to combat climate change can also be interpreted as an indication that the cultural and social changes required for comprehensive sufficiency have already begun.

For significant reduction in final energy demand and for reaching ambitious climate goals, energy sufficiency cannot be considered in isolation but must be complemented by energy efficiency measures and a rapid transition to renewable energy sources. Furthermore, the reinforcement and deployment of interconnection infrastructure is paramount to unlocking synergies between EU countries. Increased reliance on electrical backbones enables the exploitation of energy resources within Europe and reduces Europe's energy dependence to

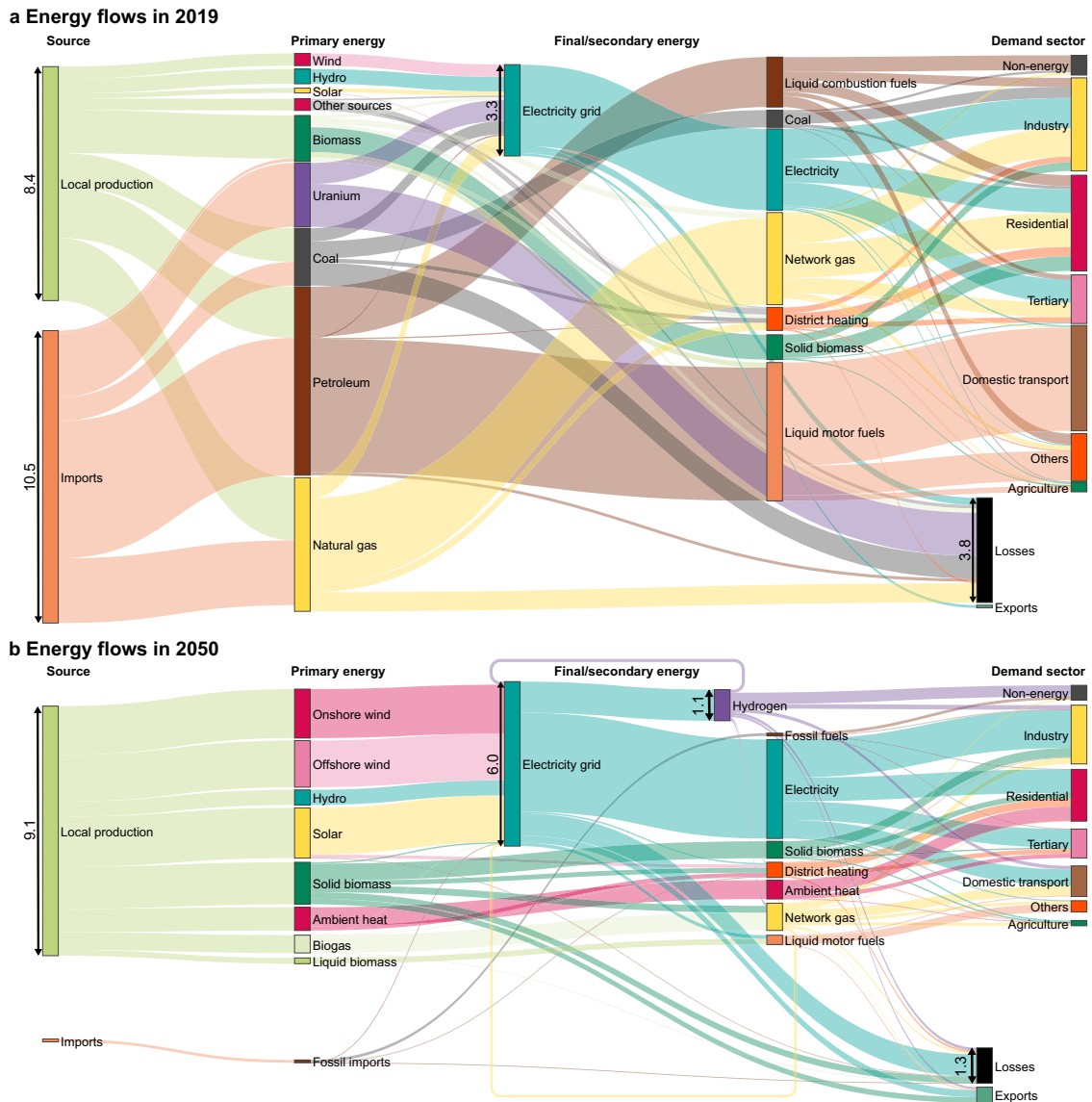

**Fig. 5 | Energy conversion chains in EU30. a** Energy flows in 2019. **b** Energy flows in 2050. The two Sankey diagrams are drawn at the same scale. Values for main flows are indicated in Petawatt hours (PWh) per year. By contrast to the rest of the document, this graph considers ambient heat, non-energy consumption, the energy sector (except blast furnaces) and maritime bunkers in the energy flows. A list of countries included in EU30 can be found in Supplementary Table 7. Source data are provided as a Source Data file.

almost zero. In the particular case of the electricity grid, a scaling up is required to host additional electricity generation capacity and meet the final electricity demand, but this increase remains limited compared to alternative European scenarios. Exchange within Europe significantly reduces the need for import of renewable fuels and externalisation of generation needs to other world regions - a key element of a global just energy transition.

However, sufficiency is a crucial lever in reducing energy demand that must be considered on a level playing field with the other mitigation strategies - efficiency and renewables. This requires an explicit inclusion of sufficiency policy in EU policy frameworks, such as the National Energy and Climate Plans and EU scenario studies. To enable an improved integration of sufficiency options in scenarios, common indicators for energy service demands, as well as the collection of statistical data for those indicators are required. To conclude, sufficiency increases the probability of reaching climate targets while significantly decreasing imports, thus lowering dependencies, risks and

the externalisation of environmental and social damage and thus qualifies - in contrast to most technical solutions - as a multi-solving strategy. It is consistent with other sustainability goals as it reduces resource needs and land use change compared to other technical solutions. As sufficiency targets imply corridors of energy use in line with well-being, it can contribute to a more just energy transition within a region and globally.

## Methods

### The Sufficiency-Efficiency-Renewables framework

The main principles underlying the CLEVER scenario are (1) GHG neutrality as soon as possible and (2) reaching 100% renewable energy supply from sources within Europe, reducing import needs to a minimum. Achieving these objectives requires a significant transformation of the energy system - both supply and demand. Thus, a whole-system, socio-technical approach was required. The Sufficiency-Efficiency-Renewables (SER) framework[22,43] was therefore adopted, which offers

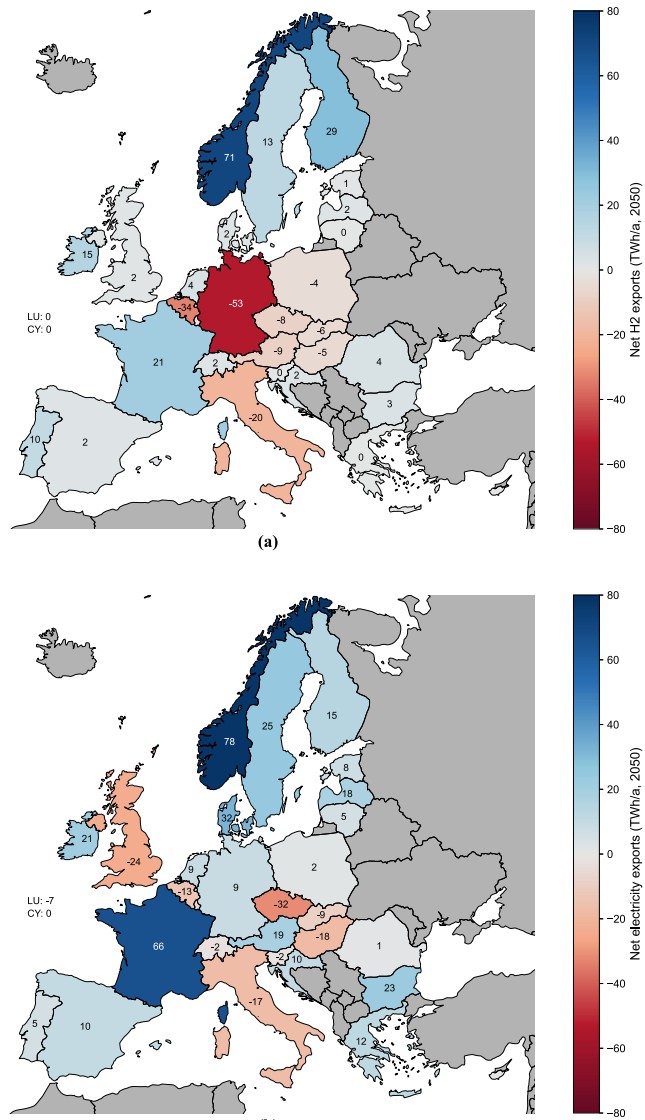

**Fig. 6 | Exchange of two main energy carriers per year within Europe in 2050 in the CLEVER (Collaborative Low Energy Vision for the European Region) scenario. a** Exchange of hydrogen. Unit: Net hydrogen (H2) export in Terawatt hours (TWh) per year (**a**). Values are rounded to the nearest TWh/a. Hydrogen trading within Europe is required for countries that have high industrial energy consumption (Germany, Italy, Belgium) and for those that have less ambitious renewable potential due to geographical constraints, such as access to the sea for offshore wind in Eastern Europe (e.g., Hungary, Slovakia and Czech Republic) or political and institutional barriers. Source data are provided as a Source Data file. **b** Exchange of electricity. Unit: Net electricity export in Terawatt hours (TWh) per year (**a**). Values are rounded to the nearest TWh/a. Transmission and trading of electricity between countries in Europe is enabled by increased levels of European political and economic collaboration. Luxembourg (LU) and Cyprus (CY) are not shown in the map, but numbers are provided. Source data are provided as a Source Data file.

demands through technical improvement, reducing losses at all stages of the energy chain. Finally, after accounting for the combined effects of sufficiency and efficiency, renewables energy sources are substituted in, displacing fossil fuels earlier and at a faster pace, thus limiting the cumulative impacts of GHG emissions.

The SER framework, with its emphasis on sufficiency on the demand-side first, allows for ambitious reductions in energy demand, as seen in the CLEVER scenario. This, in turn, allows for the scenario to achieve broader social and sustainability objectives beyond just GHG emission reductions, such as a fair distribution of mitigation efforts across countries, reducing materials throughput, and ensuring the supply-side transition can be met without relying on energy imports, or unproven and high-risk technologies.

### Greenhouse gas targets

The general objective of CLEVER is to obtain a quantified scenario that at least meets the EU targets for GHG-emissions and renewable energy shares in the years 2030 and 2050, while meeting climate-neutrality as early as possible in line with the objective to limit global warming to 1.5 degrees. In this respect, the median of C1a scenarios[44,p.20] from IPCC has been applied as the upper limit (550 GtCO$_2$) of the global carbon budget corridor and the 1.5 degree budget (500 GtCO$_2$ for 50% probability[13,p.29]) as an objective. The share of the budget assumed to be available for the EU30 countries in this study corresponds to the share in the global population (6%)[45]. For GHG-emission accounting, a territorial approach is applied: net domestic emissions are considered, excluding imported emissions of energy carriers and consumer goods. As EU targets include international flights while excluding international maritime, we apply the same accounting for comparison with EU targets. The climate neutrality and GHG emission goals apply to CO$_2$-equivalents, also including CH$_4$, N$_2$O (in agriculture) and HFC (in product uses).

### Collaborative bottom-up approach

The scenarios have been developed by 26 organisations from 20 European countries (see Bourgeois eta al.[14,p.6 and p.13] for an overview). The partner organisations are experts in energy and climate scenarios for their national countries or regions. Active partners developed and contributed national bottom-up trajectories considering country-specific circumstances. The trajectories are either based on the authors' modelling or on extraction from existing scenarios. The trajectories of countries that do not have active partners, called normalised trajectories, were built iteratively with a conservative and simplified approach.

During the first phase, for indicators indispensable for the modelling (e.g., FEC by sector and carrier, some indicators of energy production and conversion, population), basic trajectories (historical data and projection over 2019-2050) were built based on existing EU scenarios and first national scenarios gathered. Then a continuous and iterative improvement took place through: the integration of outputs from the harmonisation process, including target corridors; comparisons between improved trajectories and existing national scenarios from active partners and feedback from national commenting and observing partners.

Conversely, normalised trajectories were also good reference trajectories to inform the trajectories of active partners. Subsequently, an iterative harmonisation process took place for a consistent EU scenario. The collaborative approach enabled an active technical dialogue between the country experts and led to an exchange of best practices of energy modelling and scenario building with a focus on accounting for sustainability and sufficiency. As no standard methods exist yet for including sufficiency aspects in this kind of modelling, the dialogue and harmonisation process has set the foundation for comparing, questioning and mutually reinforcing the respective approaches.

the best opportunities for action across three cascading but transformative levers - sufficiency, efficiency and then renewables. Figure 7 provides a graphical representation of the SER framework.

Firstly, energy service levels are rescaled to a more equitable level through sufficiency measures. Sufficiency includes action at both the individual or collective level, considering both service needs and limitations (e.g., modal shifts or avoidance of unnecessary consumption). Next, sufficiency is combined with efficiency, which reduces the amount of energy required to meet the adjusted energy service

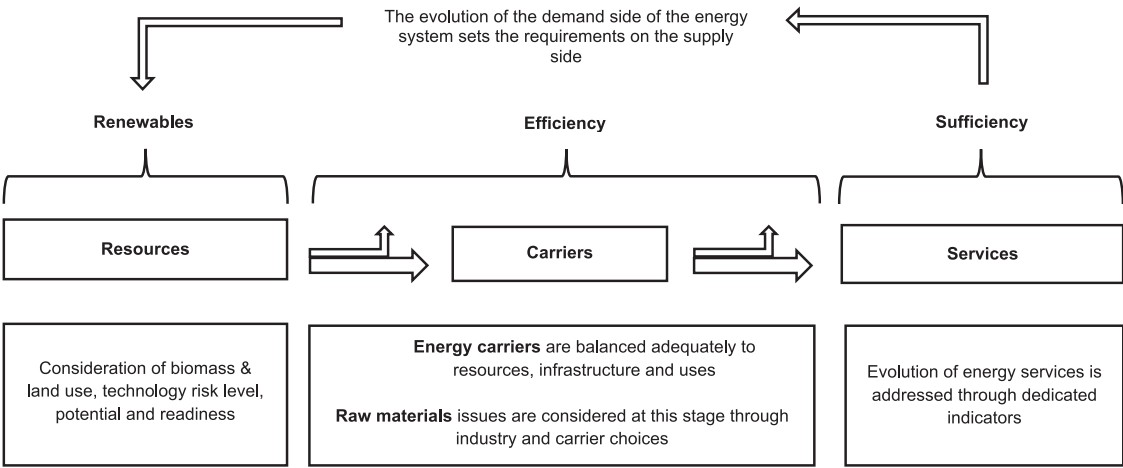

**Fig. 7 | Graphical representation of modelling framework.** The principles that underpin the model begin with a modification of energy services, based on dedicated, quantifiable indicators for each end use sector. The model then works backwards to establish the required energy level and carriers needed to meet these service demands, minimising losses throughout the energy chain. Finally, primary energy sources are matched to meet secondary and final energy demand.

## Construction of a common sufficiency modelling language

As an important precondition for constructing a harmonised sufficiency scenario, measurable key indicators on the level of energy service demand have to be defined and applied in the same way for all partners and countries. In that respect, experts from partner organisations collected and categorised sufficiency indicators and drivers. Designated working groups for the transport and building sector defined and prioritised sector-specific sufficiency indicators. Indicators were considered sufficiency-related drivers when they touch upon a dimensional (e.g., size of vehicles), service-related (intensity and duration of use of vehicles) or organisational (e.g., as the development of collective transport) characteristics of energy consumption. There are indicators of quantitative and qualitative character. The prioritisation was based on the relevance of the indicators for energy demand, their uniqueness in quantification and whether it was possible to integrate them into the model.

## Development of national trajectories

Common indicator dashboards were developed, which include the derived sufficiency indicators and other key energy system parameters. These dashboards serve as data entry files for each country and include pathways for energy demand and supply. Dashboards are structured by sectors, supply and demand and include parameters on activity levels, sufficiency indicators, technologies, efficiencies and energy carriers. The standardised template enables country comparison and serves as an interface for the modelling and commenting process. It forms the basis for translating national scenarios that are partly based on different methodologies, models, scopes and level of aggregation into a common sufficiency-focused language and data format. While a high level of detail and disaggregation for energy service demands are preferred, the dashboard also offers clustered, aggregated entry points for top-down disaggregation, where national detail was not available. With the objective to ensure a comparable starting point, indicators from Eurostat and the Odyssee-Mure database[46] were partly used. Furthermore, dedicated comparison parameters including intermediary results like final energy consumption per country were summarised. The National Calculation Model generates energy balances for each country by cross-referencing consumption and production data for each country's dashboard. Summarised dashboards per country are available under the link provided in the section Data availability below.

For the industry sector, the modelling approach differed due to the lack of detailed industry modelling per country and the interconnectedness of the sector in European countries. A top-down centralised modelling of national trajectories was undertaken guided by the options of (1) sufficiency: scaling down material demand, (2) circularity: optimise the products lifecycle and (3) efficiency: reduce the energy intensity of production. Energy-intensive sectors were modelled with higher levels of detail covering production levels, for less energy-consuming sectors an aggregated and conservative approach was taken, giving place to a narrative on industrial reshoring and the emergence of strategic industries such as green technologies as part of a reindustrialisation process ensuring employment and industrial sovereignty in Europe. The industry indicators of the CLEVER scenario are on the level of amount of material produced and are not directly calculated based on the amount of products on the service level like clothes, phones per person etc. The amounts of material produced in the CLEVER scenario are however informed by other scenarios which consider detailed analysis of consumption of goods with a high level of disaggregation, specifically the French sufficiency scenario[22]. Thus, results from detailed French scenario are adapted to the context of other European countries, which is the basis for the production corridors for other countries. The historic rate of production over consumption of material is kept in the CLEVER scenario. National partners adapted the industry trajectories according to specific national circumstances. The top-down industry trajectories are based on other major scenarios that are ambitious regarding technical options, but also in terms of circularity and sufficiency options. They thus include substantial reductions in both demand and production. An overview of the main references for the industry-sector target corridors can be found in Toledano et al.[47]. The materials considered in the CLEVER scenario include: (i) materials requiring energy-intensive industrial processes studied quantitatively on the basis of several references and corridors integrating sufficiency[47] (steel, cement, pulp and paper and chemicals and glass when data was available for countries) and (ii) other biotic or abiotic materials for non-energy uses considered qualitatively as we incorporated the assumptions and guidelines of other country/European/global-scale scenarios or studies[48–50] that indicates a substantial reduction in materials requirements under an energy transition scenario based on sufficiency (wood, crops and other foodstuffs, non-metallic ores for construction and metal products, including critical raw materials[51] such as lithium and copper.).

For the Agriculture, Forestry and Other Land Use and Bioenergy (AFOLUB) sector, the initial step has been a top-down modelling by Couturier et al.[52], which has been submitted to the national partners for revision and adjustment. The main objectives of the sector scenarios were (1) mitigation of emissions within the sector (2) substitution of fossil fuels and (3) sequestration. For this purpose, the main guiding principles were a switch to a sustainable bioeconomy (diet shift, substantial reduction in animal livestock) and 100% agroecology (a form of agriculture that supports soil conservation and organic farming). Furthermore, a cascading hierarchy of the use of biomass has been applied, from high-value products to food, buildings, and chemistry. An increase in bioenergy production was modelled thanks to products coming from agroecology practices, which do not compete with agricultural products. However, this production comes in addition to today's bioenergy production. The calculation is based on a physical model describing mass and energy flows as well as surface use also considering social and economic impacts. The core is a supply and consumption balance without cost-optimisation. Twenty-two main crops, which are 90% of agricultural land area in Europe, are modelled in detail based on EUROSTAT and FAOSTAT data while for further 100 crops aggregated information is included. The modelling approach is described in detail in Couturier et al.[52] and most relevant data and assumptions regarding bioenergy for this scenario are summarised in the Supplementary Methods.

## Harmonisation of national trajectories

A process was developed to ensure consistency in national trajectories on the level of ambition on major hypotheses. This process can be simplified in 3 steps.

First, partners were invited to collect historical and prospective data from national statistics and existing scenarios and integrate them in the common framework (dashboard).

Second, corridors, which are minimum and maximum value for a given indicator such as residential floor space per capita [m²/capita], have been developed based on (1) minimum and maximum levels in scientific literature, (2) current ranges and speed of change, (3) sector and country expertise of the partners and (4) current and expected EU policy. The basic principle behind the corridors is the convergence of living standards[53] while constraining excessive consumption associated with high-income lifestyles. For achieving adequate levels of effort, the percentage reduction in relation to a reference year has also been analysed. Further detailed descriptions of the method and the quantified corridors in the buildings and transport sector can be found in Toledano et al.[25] and Taillard et al.[24].

Third, indicators' values of initial national trajectories were compared to corridors and partners were invited to adapt values if necessary to comply with the corridors. In some cases, national specificities justified values outside of the corridors. In reality, the process has been more iterative, with input from national partners feeding into the definition or adaptation of corridors, with corridors (e.g., feedback from other countries) opening up possibilities and justifications for national partners to strengthen the ambition of the scenario. Iterative reviews for harmonisation and efforts towards compliance with EU targets also influenced the final corridors. In the case of countries without a partner able to provide feedback on the proposed values, the high corridor values have generally been retained so as not to make over-ambitious assumptions without validation from a national partner.

This methodology has been followed in particular for the residential, passenger mobility and industrial sectors, with the production of associated methodological notes. For the tertiary and freight sectors, a similar approach was followed, but with less formalisation. Some indicators were treated top-down (e.g., AFOLUB) with a partners' review to check consistency and some indicators were considered to converge in all countries (e.g., efficiency of vehicles). Some specific indicators, dependent on the national context, for example ambition on PV and wind, as described in the next chapter, were refined on a case-by-case basis.

The assumptions on population followed the EUROSTAT baseline projection[54]. Historical data (based on EUROSTAT and ODYSSEE) has been an important basis to define corridors.

## Matching supply with demand considering deep sustainability

For the harmonisation at European level, a European synthesis module was developed. The aggregated results are compared with EU targets for identifying the need for further ambition. Additionally, an energy carrier balance review is performed, in combination with additional harmonisation and convergence. This iterative process required reviews and adaptation of all national trajectories.

For each subsector, corridors for the share of each energy carrier were derived based on techno-economic evaluations. Sectoral constraints such as limited possibilities for specific industrial processes, or district heating being an option only in urbanised areas additionally limited the potentials. Guidelines from detailed materials flow modelling as well as expert consultation were applied for taking materials restrictions into account (e.g., on lithium in electric vehicles or copper for electrification). As technologies with a lower technology readiness level (TRL) than 7 (prototype is working in the expected conditions) are unlikely to be deployed at scale before 2050 or only in limited proportion, technologies with a TRL of at least 7[55] were privileged. Then partners were invited to adapt values of their national scenarios to fit into the proposed corridors.

For defining sustainable potentials of renewable energies, detailed analyses have been performed for each technology. Here we summarise the main principles. For bioenergy production, the main restrictions are food security (and more general social and societal issues), climate and biodiversity issues.

The evaluation of the sustainable bioenergy potentials based on the systemic approach described above and more in detail in the Supplementary Methods led to in total, potential domestic primary production of bioenergy in this scenario increases for EU28 from 1500 TWh in 2015 to 2290 TWh in 2050. Two thirds of the increase comes from biogas production based on manure, cover crops, some residues and biowaste. Wood energy from forest stays on the same level while other solid biomass increases due to wood-by-products, waste and agroforestry. Biofuel potential in the scenario is restricted to 150 TWh for 1st generation (slight reduction from current level), 50 TWh for 2nd generation and 20 TWh for 3rd generation. As shown in Supplementary Table 5 regarding the bioenergy potential applied in different scenarios as studies, the resulting potential of bioenergy production in CLEVER (2290 TWh) and even more the production required by the scenario (2120 TWh) are close or below several evaluations of sustainable potentials, including the threshold defined for 2040 by The European Scientific Advisory Board on Climate Change[56]. The main specificity of the CLEVER estimation is be the role of sequential cropping for biogas production, which is not included in most estimations of sustainable potential, but has the potential for simultaneous biogas and food production along with biodiversity benefits[57].

The starting point for PV and wind potentials is the techno-economic potentials identified by Ruiz Castello et al.[58]. However, country experts evaluated these country-specific potentials as very high, so they were refined on a case-by-case basis. The main reasons for difference in ambitions are the varying levels of acceptability in different countries and different circumstances for the speed of expansion. Different political status/will/announcements also contributed to adaptations in national potential estimations. In summary, the CLEVER scenario had significantly lower installed capacities in 2050 in EU27 countries than the lower variant of JRC capacities for PV (1360 GW vs 4400 GW), onshore wind (546 GW vs

2891 GW) and than the higher variant got offshore wind (328 GW vs 2710 GW).

Hydropower capacities and production have been considered stable between today and 2050. Solar thermal, ocean energies, CSP, deep geothermal and waste heat were considered but constitute rather low potentials compared to other renewable options.

As conversion technologies for Hydrogen and Power-to-X (PtX) raise sustainability issues, electrification is preferred if it presents equivalent or better overall efficiency. However, those conversion technologies are key to full decarbonisation. Priority for PtX applications is given to aviation and international shipping as well as high-temperature applications in industry and long-term electricity storage. These are the applications with high efficiency potentials and very limited decarbonisation alternatives. A threshold/maximum of 350 TWh/y of syngas production in 2050 for EU27 by biomass gasification/pyrogasification is assumed. Here, the availability of solid biomass is restricted at national level and the use of biomass in buildings, industry and district heating is prioritised, which leads to the production of 214 TWh/y of syngas.

For matching supply with demand, most critical sectors, where there are limited decarbonisation options and smaller low-carbon potentials, are supplied first. It is checked if the respective demand can be supplied with one of the suitable carriers whilst respecting the defined corridor. This process continues and ends with less critical sectors and less critical carriers, such as electricity. This modelling step includes several iterations and consistency checks, in exchange with partners. The corridors for the share of a carrier in a subsector in 2050 can be found in Bourgeois et al.[14], p.58. An overview table on the order of supply-carrier-matching can be found in Bourgeois et al.[14], p.64.

Regarding the system adequacy of the renewable electricity supply in this scenario, methane and hydrogen thermal plants, hydropower (reservoir and pumped), batteries and transmission between European countries provide flexibility. A dispatchable production corridor (in percent of the final electricity demand excluding storage and transmission) of at least 14% for each country has been determined. This results in 18% share of flexible production over total demand for EU27 including hydro reservoirs. The flexible power production in the CLEVER scenario is rather overestimated as a comparison of dispatchable production in other scenarios shows. A benchmark analysis of selected scenarios (see p.69, Table 3 in Bourgeois et al.[14]), yields a range of 8–15% flexible production, which is lower than this value in CLEVER. Transmission between European countries is generally assumed in CLEVER, but a detailed grid simulation has not been done. As exchange between countries lowers the flexibility needs within countries[59], a detailed grid modelling of the scenario would most likely emphasise that hourly dispatch would also work with lower installed capacities of flexible production than currently assumed in CLEVER.

## Limitations of this work

The proposed framework for energy systems modelling involves simplifying assumptions. Adequacy and flexibility of the system are not explicitly estimated through an hourly dispatch model, but a conservative value for flexible power generation assures the flexibility requirements. To refine this hypothesis further, a high time-resolution simulation could be used, resulting in a more accurate evaluation of the firm capacity required in the system. Similarly, the electrical grid is not explicitly modelled but assumed to be a key component of the system, and congestion between and within countries is disregarded. To describe the need for further grid reinforcement, an hourly power flow model with capacity expansion could be used. Finally, there is no explicit quantification of the investment required for new generation capacities and flexibility resources such as storage or demand response. As a result, the proposed energy mix may not correspond to a conventional economic optimum, but is the result of a robust supply-demand matching process based on expert opinion, considering the specificities of each country.

In this study, territorial-based accounting of emissions has been applied and the historic rate of production over consumption of goods and material has been kept. For a fully consistent approach referring to decent living standards also on a consumption goods level, consumption-based accounting of emissions would be required in combination. This would however imply additional model types.

While the scenario provides an illustration of the policy framework needed to enforce sufficiency measures, it does not formally include the policies within the model, nor does it explicitly quantify their potential effects. This is partly due to the lack of empirical quantitative data relating specific policy measures to the sufficiency indicators. Similarly, effects such as financial or time-use rebounds could significantly impact consumption behaviours, but they require further research and were not included in this work.

## Data availability

The input and output datasets generated during the current study are available in the Zenodo repository named Simplified Energy Prospective and Interterritorial Analysis (SEPIA) tool, applied to the CLE-VER energy transition scenario https://doi.org/10.5281/zenodo.11546125. An extract of detailed assumptions/key parameters for each country and sector as well as data on the policy instruments for all sectors considered for the scenario are provided as Supplementary Data 1 in an easy to access Excel-file. For figures within this manuscript, Source data are provided with this paper.

## Code availability

The code developed and applied for this study is available under an open license in the Zenodo repository named Simplified Energy Prospective and Interterritorial Analysis (SEPIA) tool, applied to the CLE-VER energy transition scenario https://doi.org/10.5281/zenodo.11546125, including instructions for system requirements, installation and use.

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

## Acknowledgements

F.W., J.T., C.Z.-Z. are funded by German Federal Ministry of Education and Research (BMBF), within the framework of the Strategy Research for Sustainability (FONA), as part of its Social-Ecological Research funding priority (grant numbers 01UU2004A, 01UU2004B, 01UU2004C). N.T., S.B., M.D., A.G., A.J., A.T. and Y.M. received funding in 2022 and 2023 for the CLEVER project from the European Climate Foundation (ECF) (grant number 2201-63204), from the Fondation pour le Progrès de l'Homme, from the endowment fund Watt For Change (grant number 21-49) and the French Agency for Ecological Transition (ADEME) (grant number 22ESD0042). B.M., J.C., S.M., A.R., L.P. and B.V. were in 2022 partly funded for the CLEVER project by the European Climate Foundation (ECF) (grant number 2201-63204). E.J. received funding from the Energy Demand Research Centre (EDRC) (UKRI grant award EP/Y010078/1). P.T. is funded by the Energy Transition Fund from FPS Economy, which represents the Belgian Minister of Energy, Environment and Sustainable Development. L.P. and A.R. received funding from the project FULFILL - Fundamental Decarbonisation Through Sufficiency By Lifestyle Changes, funded by the European Union's Horizon 2020 research and innovation programme (Grant Agreement No. 101003656). We acknowledge financial support by Land Schleswig-Holstein within the funding programme Open Access-Publikationsfonds.

## Author contributions

F.W. had designed, organised and structured the paper and writing process and has written main parts of the text. N.T. has coordinated the modelling work and was responsible for structuring and organising the modelling process with all national partners. Y.M. has proposed the idea of a bottom-up European scenario based on sufficiency and brought in the concept and methodological approach. N.T., F.W., Y.M., E.B., B.B., S.B., J.C., L.C., M.D., A.G., A.J., E.J., S.M., B.M., L.P., S.Q., A.R., J.T., P.T., A.T., B.V., C.Z.-Z. have been involved in the national modelling, data and figure preparation, writing and reviewing of the article.

## Funding

## Competing interests

The authors declare no competing interests.
