## [Peer Review File · Nature Communications]

Reviewers' comments:

Reviewer #1 (Remarks to the Author):

This manuscript puts together a Europewide and national-level low-demand scenario, highlighting the important role of sufficiency. This research is based on a large international project that leads to a massive and bottom-up data dashboard, thanks to the valuable stakeholder engagement and inter-institutional collaboration. The results suggest that sufficiency can be promising in massively reducing energy demand by 50% in 2050 without nuclear, CCS, or dedicated biomass crops in Europe – rather ambitious compared to the existing modeling results and literature.

I think the results are interesting in this manuscript. However, there are significant inconsistencies in data assumptions, unclear descriptions of the methods, as well as the lack of evidence to support the conclusion. Therefore, I recommend a rejection in this case. Please find the detailed comments below, considering the suggested questions by Nature Communications.

- Inconsistent data assumption/methods both within the manuscript and compared to the cited references.

The most prominent example is bioenergy and the relevant AFOLUB sector. First, there is inconsistency within the applied methods. On the one hand, the authors claim to utilize “only by-products for energy generation” (L297). On the other hand, dedicated forest biomass (like fuelwood) and dedicated crops (like oil crops) are included at the same time (as suggested by the data source ref [43]). So, what biomass is considered exactly? Still “only by-products” or actually everything?

Second, there is an obvious inconsistency between this manuscript and the existing literature cited/used by this manuscript. Looking at the “sustainable bioenergy potential” suggested by this manuscript (2250 TWh in 2050 in EU28) at L340, this number is way too high for any future bioenergy in Europe when 100% agroecology is in place (L295) and/or “only by-products for energy generation” is considered (L297). Let's do the math and look at the reference dataset that the authors are using here, the JRC ENSPRESO report [31]. It is not wrong that 2250 TWh can be the lower range of JRC bioenergy potential, INCLUDING dedicated biomass from forests and agriculture. However, when considering “only the by-products” or “100% agroecology,” as assumed in this study, there is no way for Europe to reach 2250 TWh at all. For instance, no organic agriculture/agroecology AND “only the by-products” will lead to just 1000 PJ of bioenergy in 2050 Europe (See Figure 2 “JRC (residue).”

This inconsistent assumption of bioenergy thus makes the overall results of renewables provision considerably less convincing. Since this CLEVER scenario relies quite much on bioenergy provision to reach carbon neutrality and energy security, the inconsistent assumptions of bioenergy make results questionable.

There are other inconsistencies apart from bioenergy that are considerably deviating from the current literature without sufficient scientific evidence from this manuscript. One example is L147-149, “a smaller energy system....limits both the costs....” and the 99% share of local primary energy production in 2050 without power transmission grids (L135). This conclusion is very surprising to me as a smaller and autarky energy system usually costs more than a larger one (see the graphical abstract of the European power system that demonstrates exactly the opposite conclusion – a regional-scale system would cost way much and requires more power generation even when transmission grids are available). With that being said, I cannot comprehend this result, and it does not have any further explanation/context/discussion.

- Lack of results/arguments/evidence to support the results and policy implications.

No CCS, no nuclear, no transmission, yet 100% renewable and carbon-neutral? The study says that no CCS or nuclear is needed at all for 2050, yet there are sure emissions from producing bioenergy, mobilizing biomass, transporting hydrogen, etc. My guess would be the authors assume all biomass feedstocks to be immediately “carbon-neutral” in 2050, although this is not clarified or argued in the manuscript. More importantly, what is left to balance the 100% renewable energy system in this case? There is no clue about battery/storage options, and there is no transmission capacity available. I understand a smaller energy system with lower demand requires less transmission, yet such a strong argument of absolutely no CCS or nuclear or transmission would require much better and more convincing results to support it.

- The missing Planetary Boundaries (PBs) and Critical Raw Materials (CRMs)?

PBs and CRMs are mentioned at the very beginning of the manuscript and in the methods as well (e.g., L63, L36, L42, L54...), yet they are missing in the results. For instance, the authors are saying that limiting to the other planetary boundaries is one of the “key components of the theoretical framework behind the CLEVER scenario,” yet there is nothing in the methods/results relevant to PBs. How do we make sense of this missing pillar of the CLEVER theoretical framework, then?

The same case applies to CRMs as well, which is more confusing. There are several places mentioning “materials demand,” “raw materials,” or “CRMs,” yet they seem to point to different “materials.” The most relevant explanation is at L332: “Guidelines from detailed material flow modeling as well as expert consultation were applied for taking material restrictions into account (e.g., on lithium in electric vehicles or copper for electrification).” Then what exactly are these “materials” considered in this study? And to which extent do they alter the results and the role of sufficiency, and how? “Sufficiency: scale the material demand” (L280) is the key element in this study, but it appears to be inconsistent and confusing, which leads to questionable and unclear conclusions.

- Other results/arguments/evidence that are lacking/inconsistent/confusing to support the conclusions are:

i. How do circularity and efficiency contribute to sufficiency? How do they shape the low-demand scenario together? Concurrently?

ii. PV and Wind potentials are “significantly lower than JRC” (L351) – how low? According to what kind of acceptability?

iii. Hydrogen and power-to-x are electricity-intensive. Considering the much lower PV and wind availability, is there going to be enough power left to produce green hydrogen? Or actually, the unspecified hydrogen is blue hydrogen from biomass? Then, it takes us back to the first point of whether we could really agree on the overestimation of sustainable bioenergy potential in 2050. In either case, there might very likely be a lack of hydrogen to supply carbon-neutral feedstocks, I suspect – no evidence in the study either.

iv. “Policies to support a sufficiency development” from L150. I do not think there are sufficient results to support the authors’ argument here -- effective policies are pivotal in all energy and climate scenarios (L151). I mean, I do personally agree with this argument, but there is no evidence in this study to support it. Actually, “policies were not specifically modeled, but instruments have been outlined.” I do not fully understand how outlining/listing instruments can lead to this conclusion without any analysis. This also makes the appendix Excel sheet (“Policies”) quite confusing to me.

v. Last but not least, the provided datasets and methods are not detailed or clear enough to be reproduced. E.g., How did the harmonization of data happen? To what data/country?

- Other details that make it confusing to comprehend the manuscript.

“Pkm/capita” pops up a bit out of the blue (L20). I guess it refers to transport demand. Passenger km per capita? In any case, it is confusing for me as there is no context or explanation that could directly lead to passenger km per capita. I’d suggest using the full name or just “transport demand” instead. I will not go through other details that hinder one from understanding the manuscript.

In a nutshell, considering the major inconsistency of data/methods and the lack of results supporting conclusions, I’d suggest authors considerably reorganize/rewrite/refine the manuscript. I spent quite some time figuring out the supplementary data and even the project reports, but I still cannot get a clear picture, especially the inconsistent method part. I do not think it is necessary for every reader of Nature Communications to go through all references and supplementary to grasp the main text. So, I’d suggest a rejection in this case.

Reviewer #3 (Remarks to the Author):

This manuscript presents a 1,5C-compatible 2019-2050 carbon neutrality scenario (called “CLEVER”) aligned with IPCC carbon budget and EU policy goals. The scenario is built using a bottom-up, country-level modeling approach. Its consistency and transparency are strengthened by extensive supplementary material. The novelty compared to previous scenarios is the focus on sufficiency measures in addition to energy efficiency and renewable energy technologies. As Europe is in urgent need of new solutions for achieving carbon neutrality by 2050, the paper should be of wide interest.

In my opinion, the following points need improvement:

- 1) The main article presents scenarios for two sufficiency indicators, namely “living space per capita” and “person-kilometres travelled per capita”, but the supplementary material also lists many other indicators. I would recommend listing also other sufficiency indicators that were part of the scenario in the main article to better understand what constitutes the savings through sufficiency.
- 2) How were the scenarios on the country-specific sufficiency indicators done? How was it ensured that the country-specific scenarios were consistent in approach? Table 1 lists policy strategies but it is not clear how those were considered in the country-specific scenarios.
- 3) I recommend following a standard article structure and naming the first section “Introduction”.
- 4) Strong expressions should be avoided. For example, “There is a strong scientific consensus that sufficiency habits are enabled and promoted by policy measures.” (lines 153-154) lacks evidence of a “strong scientific consensus” and could be rephrased as “Previous research suggests...”.

Thank you very much for reviewing. Your comments have helped us a lot to substantially improve the manuscript.

In the following we answer each of the reviewers' comments in blue.

Reviewer #1 (Remarks to the Author):

This manuscript puts together a Europe-wide and national-level low-demand scenario, highlighting the important role of sufficiency. This research is based on a large international project that leads to a massive and bottom-up data dashboard, thanks to the valuable stakeholder engagement and inter-institutional collaboration. The results suggest that sufficiency can be promising in massively reducing energy demand by 50% in 2050 without nuclear, CCS, or dedicated biomass crops in Europe – rather ambitious compared to the existing modeling results and literature.

1.0

I think the results are interesting in this manuscript. However, there are significant inconsistencies in data assumptions, unclear descriptions of the methods, as well as the lack of evidence to support the conclusion. Therefore, I recommend a rejection in this case. Please find the detailed comments below, considering the suggested questions by Nature Communications.

ANSWER: Thank you very much for the thorough review. We have deeply discussed all comments and conclude that we have not formulated our manuscript precisely enough, which has led to several misunderstandings. We would like to clear up such misunderstandings and have thus answered in detail to each of your comments in the following. We agree with many of your comments and agree with the need to be more clear and precise in formulation. We have thus thoroughly revised the text. We provide one version of the manuscript with marked changes and one clear one.

Additionally, we realised thanks to your comments, that almost all background information is available but partly difficult to discover for the reader in all the referenced documents. We have thus also added supplementary material to distil the most relevant information for the reader without breaking the strict word maximum of Nature articles.

We very much hope that this will clear up any misunderstandings and convince you of the scientific quality of our research and the novel value of the results.

1.1

- Inconsistent data assumption/methods both within the manuscript and compared to the cited references.

The most prominent example is bioenergy and the relevant AFOLUB sector. First, there is inconsistency within the applied methods. On the one hand, the authors claim to utilize “only by-products for energy generation” (L297). On the other hand, dedicated forest biomass (like fuelwood) and dedicated crops (like oil crops) are included at the same time (as suggested by

the data source ref [43]. So, what biomass is considered exactly? Still “only by-products” or actually everything?

Second, there is an obvious inconsistency between this manuscript and the existing literature cited/used by this manuscript. Looking at the “sustainable bioenergy potential” suggested by this manuscript (2250 TWh in 2050 in EU28) at L340, this number is way too high for any future bioenergy in Europe when 100% agroecology is in place (L295) and/or “only by-products for energy generation” is considered (L297). Let’s do the math and look at the reference dataset that the authors are using here, the JRC ENSPRESO report [31]. It is not wrong that 2250 TWh can be the lower range of JRC bioenergy potential, INCLUDING dedicated biomass from forests and agriculture. However, when considering “only the by-products” or “100% agroecology,” as assumed in this study, there is no way for Europe to reach 2250 TWh at all. For instance, no organic agriculture/agroecology AND “only the by-products” will lead to just 1000 PJ of bioenergy in 2050 Europe (See Figure 2 “JRC (residue).”

This inconsistent assumption of bioenergy thus makes the overall results of renewables provision considerably less convincing. Since this CLEVER scenario relies quite much on bioenergy provision to reach carbon neutrality and energy security, the inconsistent assumptions of bioenergy make results questionable.

ANSWER:

For the bioenergy potential, a complete modelling was done, which takes into account the trade offs and conflicts regarding bioenergy use, agriculture and carbon sequestration.

The AFOLUB scenario is based on a physical model describing mass and energy flows as well as surface use also considering social and economic impacts. The core of the model is a supply and consumption balance without cost-optimisation. 22 main crops, which correspond to 90% of agricultural land area in Europe, are modelled in detail based on EUROSTAT and FAOSTAT data while for further 100 crops aggregated information is included.

This modelling goes into much more detail than for other energy scenarios, especially with the integration of agroecology practices.

We think that your assumption on inconsistency of our bioenergy assumptions mainly comes from different understandings of the term “by-products”. We have adapted the manuscript avoiding this term, better describing that the additional bioenergy potential is a result of a detailed analysis of changing agricultural practices and a more sustainable management of the land (in line with the idea of the scenario to also keep other planetary boundaries in mind). CLEVER does not use bioenergy to a greater extent than other scenarios. We have adapted the two paragraphs on bioenergy in the method section (L323-338 and L387-400) to better explain our approach and have added supplementary material (Supplementary1_BioenergyPotential.pdf) with more but precise background also providing a comparison to other studies.

1.2.

There are other inconsistencies apart from bioenergy that are considerably deviating from the current literature without sufficient scientific evidence from this manuscript. One example is

L147-149, “a smaller energy system....limits both the costs....” and the 99% share of local primary energy production in 2050 without power transmission grids (L135). This conclusion is very surprising to me as a smaller and autarky energy system usually costs more than a larger one (see the graphical abstract of the European power system that demonstrates exactly the opposite conclusion – a regional-scale system would cost way much and requires more power generation even when transmission grids are available). With that being said, I cannot comprehend this result, and it does not have any further explanation/context/discussion.

ANSWER:

Reviewer #1 concludes that we consider autarkic systems without transmission systems. This is a fundamental misunderstanding. Our approach is to use **local** sources as much as possible, but an exchange via transmission grids between countries within Europe is an essential characteristic and exchanges of energy between countries are highlighted in the scenario. We agree that autarkic systems are more costly. By smaller system we mean that less renewable supply-side infrastructure is required to displace fossil fuel consumption. High reductions in energy demand also result in fewer residual emissions in 2050, meaning that expensive carbon capture technologies do not inflate investment requirements (see Barrett et al., 2022). We reformulated the text in several places to avoid misunderstanding. We have also adapted the wording “local” and opted to better explain the cost claim, whilst moving it to the discussion section.

- In the introduction we have added the words in red: “In this paper, we present a scenario at the European level with a strong focus on sufficiency options while strongly reducing dependency on energy imports from outside Europe, enabling exchange and transmission of hydrogen and power between European countries as an essential characteristic,...
- We have included a new figure (6b), which shows the exchange of power between European countries, to better demonstrate the high exchange of electricity between countries
- We have deleted the sentence “a smaller energy system ... limits both the costs ...” and have instead added a paragraph (L171-180) in the discussion section explaining why we assume that a smaller energy system potentially reduces costs.

1.3

- Lack of results/arguments/evidence to support the results and policy implications.

No CCS, no nuclear, no transmission, yet 100% renewable and carbon-neutral? The study says that no CCS or nuclear is needed at all for 2050, yet there are sure emissions from producing bioenergy, mobilizing biomass, transporting hydrogen, etc. My guess would be the authors assume all biomass feedstocks to be immediately “carbon-neutral” in 2050, although this is not clarified or argued in the manuscript. More importantly, what is left to balance the 100% renewable energy system in this case? There is no clue about battery/storage options, and there is no transmission capacity available. I understand a smaller energy system with lower demand requires less transmission, yet such a strong argument of absolutely no CCS or nuclear or transmission would require much better and more convincing results to support it.

ANSWER:

Carbon neutrality of Biomass: Bioenergies used for energy purposes in this scenario are considered as being carbon neutral on a short cycle, in the order of a year, as the carbon stock in bioenergies neither increases nor decreases.

What is left to balance the system: Transmission grid, different storages including batteries as well as flexible power production are available for balancing. So far, the balancing has been described as follows in the paper: "To ensure the system adequacy of the renewable electricity supply in this scenario, a dispatchable production corridor (in percent of the final electricity demand excluding storage) of 14% has been determined based on other studies (see [15, p.69] for a list of references and values). The value has been applied for each country. Thus, the flexible power production required is rather overestimated as exchange between countries would lower the flexibility needs within countries [49]." We have reformulated to better explain the options, also naming the options within the manuscript, also stressing that assumptions regarding dispatchable capacity are on the conservative side in comparison to other studies. We also emphasise once again, that transmission is possible in the scenarios. The reformulated paragraph is in L427-438.

CCS and nuclear could add flexibility only to a limited extent to a system, as those processes do not allow fast start-up and shutdown. Furthermore, several publications describe options for balancing 100% renewable systems without CCS and without nuclear power, like e.g. Jacobson et al. (2018) [<https://doi.org/10.1016/j.renene.2018.02.009>] listing 24 advanced studies that have examined matching time-dependent demand with supply for up to 100% renewable energy.

According to Breyer et al. [10.1109/ACCESS.2022.3193402], electricity scenarios without fossil and/or nuclear fuels has sparked criticism (e.g., Clack [10.1073/pnas.1610381114]; Heard et al. [10.1016/j.rser.2017.03.114]) with some authors putting doubt on the technical feasibility of renewable-based electricity systems. However, for Breyer et al., many of those criticisms were not sustained when examined in detail by an increasing number of studies, particularly after 2017. Studies that addressed such technical concerns include Aghahosseini et al. [10.1016/j.rser.2019.01.046], Jacobson et al. [10.1016/j.renene.2021.11.067; 10.1073/pnas.1510028112; 10.1073/pnas.1708069114], and Sgouridis et al. [10.1016/j.erss.2022.102497] who directly addressed Clack [10.1073/pnas.1610381114] and the work of Brown et al. and Diesendorf et al. [10.1016/j.rser.2018.05.042] who directly responded to the critics by Heard et al.

Most of the criticism is based on renewable sources being variable. However, there is a range of strategies for dealing with variability. A combination of strategies such as demand response [10.1016/j.apenergy.2015.10.083], expansion of connection to neighbouring countries [10.1016/j.renene.2021.07.115; 10.1016/j.renene.2013.10.005], storage [10.1016/j.joule.2020.11.015], sector coupling [10.1016/j.energy.2012.04.003.], and power-to-x [10.3390/en14206594] can mitigate the variability problem.

1.4

- The missing Planetary Boundaries (PBs) and Critical Raw Materials (CRMs)?

PBs and CRMs are mentioned at the very beginning of the manuscript and in the methods as well (e.g., L63, L36, L42, L54...), yet they are missing in the results. For instance, the authors are saying that limiting to the other planetary boundaries is one of the “key components of the theoretical framework behind the CLEVER scenario,” yet there is nothing in the methods/results relevant to PBs. How do we make sense of this missing pillar of the CLEVER theoretical framework, then?

ANSWER:

Thank you for this comment, indeed we had not expanded on the impact of our scenario on other planetary boundaries due to space constraints. We agree that this could be confusing for the reader to understand our modelling choices and the principle of deep sustainability that underpins them.

To remedy this, we have now prepared additional supplementary material for this article that provides insights into the qualitative and quantitative impacts of our scenario on the planetary boundaries that we used to guide our choices (Supplementary3_PlanetaryBoundaryAspects). This can also better explain our strategy for the bioenergy trajectory, which has impacts on several planetary boundaries through the links to agricultural choices.

Our answer to your comment regarding CRMs is summarised in our answer to your next comment that is more specifically on CRMs.

1.5

The same case applies to CRMs as well, which is more confusing. There are several places mentioning “materials demand,” “raw materials,” or “CRMs,” yet they seem to point to different “materials.” The most relevant explanation is at L332: “Guidelines from detailed material flow modeling as well as expert consultation were applied for taking material restrictions into account (e.g., on lithium in electric vehicles or copper for electrification).” Then what exactly are these “materials” considered in this study? And to which extent do they alter the results and the role of sufficiency, and how? “Sufficiency: scale the material demand” (L280) is the key element in this study, but it appears to be inconsistent and confusing, which leads to questionable and unclear conclusions.

ANSWER:

Thank you for these questions. The CLEVER scenario incorporates an overall reduction in materials in energy-intensive industrial sectors, in line with the sufficiency approach. A quantitative analysis (shown in Figure 4), based on various references and corridors for production and consumption (CLEVER 2022), focuses on steel, cement, paper pulp and, in specific cases where data is available, on the consumption of petroleum products for olefin production and Sicilian sand for glass production. These sectors accounted for around 55% of final energy consumption in the European industrial sector in 2019 (EUROSTAT 2023), and the scenario shows that the sufficiency measures applied to reduce energy consumption in the building and transport sectors in particular (e.g., reducing the surface area of buildings per

inhabitant, reducing the size of cars or shifting the modal shift from car to bicycle and public transport) also lead to a reduction in demand for these materials.

Other studies (Rauzier and Toulouse 2022, Cabeza et al. 2022, Transport and Environment 2023) focus on how sufficiency and demand reduction assumptions lead to a reduction in various other materials on a global, European and national scale. These include wood, foodstuffs, textiles, non-metallic ores for construction and metal products, including critical raw materials (CRM) (European Commission 2023) such as lithium and copper. Research conducted by Rauzier and Toulouse (2022) on a French scale indicates a substantial reduction in material requirements under an energy transition scenario based on the sufficiency-efficiency-renewables framework. These results and the assumptions considered have been used as guidelines in the development of the CLEVER scenario, to integrate these issues in a qualitative way. The correlation observed in the literature between energy sufficiency and indirect impacts on materials demand suggests that the CLEVER scenario should therefore result in an overall reduction in the consumption of all materials although calculations have not been carried out for every material. Regarding CRMs for transport electrification, for example, we ensured that CLEVER's transport assumptions, in particular the reduction in per capita car travel and vehicle size, were close to those considered by Rauzier, Toulouse (2022). Their study indicates a substantial reduction in lithium, cobalt and copper consumption compared with a Business-As-Usual (BAU) road transport electrification scenario without sufficiency.

The CLEVER scenario, driven by sufficiency, mitigates material-related pressures, resulting in a reduction in environmental impacts integrated into other planetary boundaries than climate change, including land system change, freshwater change and biosphere integrity. This is particularly relevant for metals, as shown by the International Resource Panel (2019), where increased production, due to reduced grades, leads to an increase in mining impacts according to the OECD (2019). In addition, the reduction in demand for materials contributes to reducing bottlenecks in the supply of critical materials, particularly those needed for electric batteries (Seck et al. 2022, European Court of Auditors, 2023).

In addition, Watari et al. (2020) and International Energy Agency (2019) show that mitigating climate change requires more efficient production, as well as a reduction in the production of metals. This strengthens the link between the mitigation of climate change through decreased energy consumption and the diminution of the demand for metals.

To mitigate confusion arising from different terminologies, we have revised the manuscript to use the term "materials" exclusively. The first occurrence of this term includes a description of what is meant by the demand for materials, ("including (i) materials requiring energy-intensive industrial processes and (ii) other biotic or abiotic materials for non-energy uses". Additionally, we have added information in the method section specifying the perimeter considered, distinguishing between materials studied quantitatively and those assessed qualitatively.

Associated references:

CLEVER, 2022: Establishment of energy consumption convergence corridors to 2050 Industrial sector - Industry sector.

<https://clever-energy-scenario.eu/wp-content/uploads/2023/02/2206-Convergence-corridors-Industry.pdf>

Eurostat, 2023: Complete energy balances (updated 28/04/2023)

https://ec.europa.eu/eurostat/databrowser/view/nrg_bal_c__custom_9514295/default/table?lang=en

Rauzier, E. and Toulouse, E., 2022: The material impacts of an energy transition based on sufficiency, efficiency, and renewables. ECEEE SUMMER STUDY PROCEEDINGS.

https://www.negawatt.org/IMG/pdf/220608_ecee_the-material-impacts-of-an-energy-transition-based-on-sufficiency-efficiency-and-renewables.pdf

Cabeza, L. F., Q. Bai, P. Bertoldi, J.M. Kihila, A.F.P. Lucena, É. Mata, S. Mirasgedis, A. Novikova, Y. Saheb, 2022: Buildings. In IPCC, 2022: Climate Change 2022: Mitigation of Climate Change. Contribution of Working Group III to the Sixth Assessment Report of the Intergovernmental Panel on Climate Change. Cambridge University Press, Cambridge, UK and New York, NY, USA.

https://www.ipcc.ch/report/ar6/wg3/downloads/report/IPCC_AR6_WGIII_Chapter09.pdf

Transport & Environment, 2023: Clean and lean: Battery metals demand from electrifying passenger transport.

<https://www.transportenvironment.org/wp-content/uploads/2023/07/Battery-metals-demand-from-electrifying-passenger-transport-2.pdf>

European Commission, 2023: Study on the critical raw materials for the EU 2023 – Final report. Publications Office of the European Union.

<https://op.europa.eu/en/publication-detail/-/publication/57318397-fdd4-11ed-a05c-01aa75ed71a1>

International Resource Panel, 2019: Global Resources Outlook 2019: Natural Resources for the Future We Want. A Report of the International Resource Panel. United Nations Environment Programme. Nairobi, Kenya.

<https://www.resourcepanel.org/reports/global-resources-outlook>

OECD, 2018: Global Material Resources Outlook to 2060: Economic Drivers and Environmental Consequences, OECD Publishing, Paris.

<https://doi.org/10.1787/9789264307452-en>

Seck, G.,S., Hache, E., Barnet, C., 2022: Potential Bottleneck in the Energy Transition: the Case of Cobalt in an Accelerating Electro-Mobility World. Resources Policy, 2022, 75, pp.102516. <https://doi.org/10.1016/j.resourpol.2021.102516>

European Court of Auditors, 2023: Special report - The EU's industrial policy on batteries. New strategic impetus needed.

https://www.eca.europa.eu/ECAPublications/SR-2023-15/SR-2023-15_EN.pdf

Watari, T., Nansai, K., Giurco, D., Nakajima, K., McLellan, B., Helbig, C., 2020: Global

Metal Use Targets in Line with Climate Goals. *Environmental Science & Technology, Environ. Sci. Technol.* 2020, 54, 19, 12476–12483.

<https://doi.org/10.1021/acs.est.0c02471>

International Energy Agency, 2019: Material efficiency in clean energy transitions.

https://iea.blob.core.windows.net/assets/52cb5782-b6ed-4757-809f-928fd6c3384d/Material_Efficiency_in_Clean_Energy_Transitions.pdf

1.6

• Other results/arguments/evidence that are lacking/inconsistent/confusing to support the conclusions are:

i. How do circularity and efficiency contribute to sufficiency? How do they shape the low-demand scenario together? Concurrently?

ANSWER:

Thank you for this question. Our research is actually based on a very detailed analysis of sufficiency, efficiency and circularity potentials.

The underlying approach of the low-demand scenario is the Sufficiency-Efficiency-Renewables (SER) framework. Figure SM.1 in the supplementary information provides a graphical representation of the underlying principle. The first step of the modelling is the energy service level. Through dedicated indicators like e.g. m² living space or person-km the potential reduction of energy service demand in all sectors is calculated (sufficiency). Then, efficiency options are assessed. Efficiency measures reduce the level of resources required to serve the level of energy service required by improving technical performances (reducing losses) at all stages of processing. This final energy demand is then matched with renewable resources considering potentials, readiness and technology risk level. Several iteration steps have been involved. Sufficiency and efficiency do complement each other for reaching the high levels of demand reduction. 50% of final energy demand reduction is realised, of which at least 40% are due to sufficiency (as stated in the abstract and L36-37 of the paper).

In Figure 4 of the paper, we have provided numbers for contribution to reductions by sufficiency and efficiency+circularity for three industry sectors that are highly relevant regarding energy demand and GHG-emissions in Europe: Steel, Cement, Pulp&paper.

Circularity could be considered as energy efficiency (reducing unit energy consumption) or sufficiency (reducing the primary raw materials demand). For the case of steel, efficiency was considered as improving the energy intensity of the same process, circularity as switching from primary to recycled steel leading to energy demand reduction as the recycled process is less energy intensive, and sufficiency as reducing the total steel production.

1.7

ii. PV and Wind potentials are “significantly lower than JRC” (L351) – how low? According to what kind of acceptability?

ANSWER:

We have included the following text to the methods section of the manuscript for clarity (L405-408):

“The CLEVER scenario had significantly lower installed capacities in 2050 in EU27 countries than the lower variant of JRC capacities for PV (1360GW vs 4400GW), onshore wind (546GW vs 2891GW) and than the higher variant got offshore wind (328GW vs 2710GW).”

The Sankey diagram in Figure 5 shows the source that provides the flows of the whole energy system. 1.1 PWh/y of green hydrogen are produced by renewable energy sources, namely wind, solar, hydro. It is possible to provide enough solar and wind energy for direct electricity use as well as for hydrogen generation due to the much lower demand for energy and especially hydrogen than in other scenarios.

1.8

iii. Hydrogen and power-to-x are electricity-intensive. Considering the much lower PV and wind availability, is there going to be enough power left to produce green hydrogen? Or actually, the unspecified hydrogen is blue hydrogen from biomass? Then, it takes us back to the first point of whether we could really agree on the overestimation of sustainable bioenergy potential in 2050. In either case, there might very likely be a lack of hydrogen to supply carbon-neutral feedstocks, I suspect – no evidence in the study either.

ANSWER:

We provide evidence on hydrogen feedstocks in the article, illustrated in the Sankey diagram of energy flows (Figure 5). As per the above, all of the hydrogen produced in the CLEVER scenario in 2050 is green hydrogen, produced using renewable sources of electricity. 6 PWh/y of electricity is produced in 2050, of which 1.1 PWh is used in hydrogen production. This leaves 100% of biomass production to be used for other purposes, such as direct industrial demand, residential heating, or transport fuels. Again, this is possible because of the low hydrogen consumption within the scenario.

1.9

iv. “Policies to support a sufficiency development” from L150. I do not think there are sufficient results to support the authors’ argument here -- effective policies are pivotal in all energy and climate scenarios (L151). I mean, I do personally agree with this argument, but there is no evidence in this study to support it. Actually, “policies were not specifically modeled, but instruments have been outlined.” I do not fully understand how outlining/listing instruments can lead to this conclusion without any analysis. This also makes the appendix Excel sheet (“Policies”) quite confusing to me.

ANSWER:

Policies to support a sufficiency development

Thank you for your comment regarding the policy strategies. Your comment made clear that we need to better explain the role the policies strategies play in the scenario.

We do not provide direct evidence from this study that “effective policies are pivotal in all energy and climate scenarios”. We wrote this sentence in the manuscript to explain why we put a lot of effort in backing up the assumptions with policies. We actually do not know any energy and climate scenarios that explicitly and endogenously model all policies and their quantitative impact in such scenarios. Instead, we propose policies that shift framework conditions to realise the assumptions taken in this scenario to support the sufficiency assumptions in particular, by indicator and sector respectively. Many climate-neutrality scenarios modelling mostly renewables and efficiency do not back their assumptions by listing policies that lead to development assumed in the scenario, while some do provide accompanying narrative policy scenarios. We take this further step and include it in our research. However, we do not provide empirical evidence for a direct connection between policy and quantified demand reduction, since to our knowledge there is a profound research gap that we cannot close with this study. We have explained this now better in the text. Also, we have moved the policy part to the discussion section to prevent any confusion regarding the causality of policies and modelling assumptions (L181-200).

1.10

v. Last but not least, the provided datasets and methods are not detailed or clear enough to be reproduced. E.g., How did the harmonization of data happen? To what data/country?

ANSWER:

The harmonisation of data between countries was mainly carried out following the process related to indicators' corridors. We have adapted the paragraph “Harmonisation of national trajectories” in the updated manuscript for an improved description with more details of the harmonisation process (L339-369). This process has been used in particular for indicators where national differences can be the strongest like carriers' share in subsectors and sufficiency indicators.

Regarding the provision of datasets / model and the reproducibility, we additionally provide main parameters of the scenario in additional supplementary material CLEVER_data.xlsx. Here main parameters for each sector and country can be found. Furthermore, we are happy to disclose the full model including all datasheets and a short description on how to apply it under the following link: <https://seafiler.institut-negawatt.com/f/4ffbc313561f42f1941c/?dl=1>

1.11

- Other details that make it confusing to comprehend the manuscript.

“Pkm/capita” pops up a bit out of the blue (L20). I guess it refers to transport demand.

Passenger km per capita? In any case, it is confusing for me as there is no context or explanation that could directly lead to passenger km per capita. I'd suggest using the full name

or just “transport demand” instead. I will not go through other details that hinder one from understanding the manuscript.

ANSWER:

Thank you for spotting this. We have now adapted the text to better explain this unit in the caption of Figure 3b: “In the following we use the unit pkm/cap which corresponds to the average km travelled per person in a year”. We have also replaced “traffic” by “transport demand” as correctly suggested by you, but prefer to continue using the exact name of the unit, as passenger-kilometres more specifically defines the indicator. We have also checked that other units in the manuscript are sufficiently described and understandable. However, we would be pleased if you could name the other details you mention that hinder you from understanding the manuscript, in order to be able to improve this.

1.12 - summary

In a nutshell, considering the major inconsistency of data/methods and the lack of results supporting conclusions, I’d suggest authors considerably reorganize/rewrite/refine the manuscript. I spent quite some time figuring out the supplementary data and even the project reports, but I still cannot get a clear picture, especially the inconsistent method part. I do not think it is necessary for every reader of Nature Communications to go through all references and supplementary to grasp the main text. So, I’d suggest a rejection in this case.

ANSWER:

With our detailed answers above, we hope that we have clarified our approach/methods and have cleared up the misunderstandings to prove that there are no inconsistencies in our work. Your comments are very valuable for us, but we assert that our method and data are sound. However, we concede that we need to describe our research in a way that it is understandable for a broad readership and that it is possible to understand all the components of such a complex work. We have thus deeply reworked our manuscript as you can see in the clean and version with track changes and have added additional supplementary information. Main results and information are still in the main text, but as it has been such a large research project, we have realised the need to provide more extensive and precise information on some areas.

Therefore, we have included the following documents as part of the Supplementary Materials:

- Supplementary1_Data_CLEVER.xlsx: Extract of detailed assumptions / key parameter for each country and sector
- Supplementary2_BioenergyPotential.pdf: Description of the bioenergy potential applied in the scenario including modelling approach, resulting numbers and comparison to other sources
- Supplementary3_PlanetaryBoundaryAspects.pdf: Description to which extent and how other planetary boundaries than climate change have been considered in and were affected by the CLEVER scenario
- Supplementary4_Policies.xlsx: Database of policy instruments for all sectors considered for the scenario including a policy strategy overview and instrument type evaluation

Reviewer #3 (Remarks to the Author):

This manuscript presents a 1,5C-compatible 2019-2050 carbon neutrality scenario (called “CLEVER”) aligned with IPCC carbon budget and EU policy goals. The scenario is built using a bottom-up, country-level modeling approach. Its consistency and transparency are strengthened by extensive supplementary material. The novelty compared to previous scenarios is the focus on sufficiency measures in addition to energy efficiency and renewable energy technologies. As Europe is in urgent need of new solutions for achieving carbon neutrality by 2050, the paper should be of wide interest.

Thank you very much for this assessment. Your comments helped a lot to further improve our manuscript. Below please find detailed answers to your comments and descriptions of how these improvements have been incorporated into the text and Supplemental Material.

In my opinion, the following points need improvement:

2.1

1) The main article presents scenarios for two sufficiency indicators, namely “living space per capita” and “person-kilometres travelled per capita”, but the supplementary material also lists many other indicators. I would recommend listing also other sufficiency indicators that were part of the scenario in the main article to better understand what constitutes the savings through sufficiency.

ANSWER:

This is a very good point. We have added a table (new Table 1) in the main text providing an overview of main sufficiency indicators applied in CLEVER. Additionally, in the additional Supplementary Material “Supplementary1_Data_CLEVER.xlsx” we provide an extract of detailed assumptions / key parameters for each country and sector including data for main sufficiency indicators.

2.2

2) How were the scenarios on the country-specific sufficiency indicators done? How was it ensured that the country-specific scenarios were consistent in approach? Table 1 lists policy strategies but it is not clear how those were considered in the country-specific scenarios.

ANSWER:

The consistency of country-specific scenarios was mainly carried out following the process related to indicators’ corridors. We have adapted the paragraph “Harmonisation of national trajectories’ ” in the updated manuscript for an improved description with more details of the harmonisation process. This process has been used in particular for indicators where national differences can be the strongest like carriers’ share in subsectors and sufficiency indicators.

Furthermore, we are happy to disclose the full model including all datasheets and a short description on how to apply it under the following link:

<https://seafile.institut-negawatt.com/f/4ffbc313561f42f1941c/?dl=1>

Policies to support a sufficiency development

Thank you for your comment regarding the policy strategies. Your comment made clear that we need to better explain the role the policies strategies play in the scenario.

Since effective policies are pivotal to realise the goals described in all energy and climate scenarios we are backing up the assumptions with policies, but we do not provide empirical evidence for a direct connection between policy and quantified demand reduction, since to our knowledge there is a profound research gap that we cannot close with this study. To our knowledge, there is yet too little empirical evidence for how much reduction which policy would lead to as not many policies of that kind are in place yet. Thus, a broad set of policies has been researched partly being already in place, partly being proposed in National Energy and Climate Plans or by Citizen Assemblies on Climate Change or found in scientific literature. From that policies have been chosen, which resulted in the extensive table that can be found in the supplementary material on policies (Supplementary4_Policies.xlsx). These policies were one aspect the country experts discussed and had in mind for determining the indicators for the country-wise modelling. Although no direct quantification was done, the policies provided an additional orientation to determine the indicators.

Thus, summarising, we propose policies that shift framework conditions to realise the assumptions taken in this scenario to support the sufficiency assumptions in particular, by indicator and sector respectively. Many climate-neutrality scenarios modelling mostly renewables and efficiency do not back their assumptions by listing policies that lead to development assumed in the scenario, while some do provide accompanying narrative policy scenarios. We take this further step and include it in our research.

We have explained this now better in the text. Also, we have moved the policy part to the discussion section to prevent any confusion regarding the causality of policies and modelling assumptions (L181-200). The overview Table on main policy strategies (Table1 in the former version of the manuscript) and the instrument type figure (Figure 7 in the former version of the manuscript) have been moved to the supplementary material on policies.

2.3

3) I recommend following a standard article structure and naming the first section "Introduction".

ANSWER:

Thank you. We have adapted the section headings accordingly.

2.4

4) Strong expressions should be avoided. For example, "There is a strong scientific consensus that sufficiency habits are enabled and promoted by policy measures." (lines 153-154) lacks evidence of a "strong scientific consensus" and could be rephrased as "Previous research suggests...".

ANSWER:

Thank you very much for pointing this out. We have completely rewritten the paragraph with the strong expression you mention, due to the need for more clearly describing the policy part of the scenario. Additionally, we have thoroughly checked the text for these strong expressions and have adapted them accordingly in several places in the text.

REVIEWER COMMENTS

Reviewer #3 (Remarks to the Author):

Dear authors,

Thanks for your careful efforts to improve the manuscript. The revised version appropriately clarifies my doubts and is much more clear than the original submission. Well done!

The manuscript would still benefit from proof-reading to correct the spelling mistakes (e.g. L48 “Deliverontribution”, L61 “mproved”, L362 “he calculation”), unclear sentences (e.g. INTRODUCTION L36-38. “50% of final energy demand reduction is realised (compared to 2019), of which at least 40% are due to sufficiency.” 40% reduction of “final energy demand” or “final energy demand reduction”?) and language and choice of words (e.g. L161-162 “energy imports from outside of Europe are almost entirely reduced”. Replace “reduced” by some other word.).

Reviewer #4 (Remarks to the Author):

The paper successfully addresses key aspects such as methodology, results, implications, and contributions to the field, making it suitable for publication.

Reviewer #5 (Remarks to the Author):

This is in my opinion an important and timely contribution to the climate mitigation literature. It explores (thoroughly) a fair and feasible pathway for Europe to live up to its climate responsibilities and devotes the necessary attention to sufficiency alongside the more traditional approaches’ focus on clean energy and efficient technologies. I would be happy to see it published.

I have two major comments, which can be addressed via discussion and do not necessitate further analysis, and various other minor comments.

Major comments:

1. My first major comment involves the use of Decent Living Standards (DLS). These feature quite heavily in the paper's introduction, but it is not entirely clear to me how they enter into the authors' scenario. There are two separate issues here:

One is more minor, and it is that I am not sure how DLS relate to the bottom of the corridors for residential floor space and mobility shown on Figure 3? The values for residential floor space in the DLS literature are certainly much lower than the ~35 m²/cap shown on Figure 3a.

The second issue is that much of the DLS inventory refers to goods and services at the product level – phones and laptops, clothes and washing facilities – or services for which Europe relies upon substantial importation of products – i.e., healthcare. If I understand correctly, the current work instead only looks at industry with respect to production of key materials – e.g., steel, cement, paper. A proper DLS analysis, then, requires analysis of specific products, but this is very hard to do thoroughly with a large full-economy model as the authors use. I would thus suggest instead adding this important point to the discussion somewhere (perhaps briefly in the main text and in more detail in the limitations section of the methods?)

The other consequent implication that deserves mentioning is that while the scenario the authors develop may avoid European imports of energy, is it unlikely to avoid imports of manufactured goods – e.g., vehicles, home appliances and electronics, clothing, cleaning products. So there remains potential justice issues, as such imports may continue to happen through unequal exchange, if Europe retains its current purchasing power advantages over the Global south, which allow the exploitation of low-wage labour via globalised supply chains. Lines 159-161 may be a good place to highlight this.

2. The second major comment I have is that there is no consideration of inequalities within countries, from what I can see. Sufficiency floors in terms of m²/cap of living space or pkm of mobility, must apply to individuals to be meaningful, and a national average exceeding a sufficiency floor does not guarantee decent living standards are available to all. Considering Figure 3, I suspect the authors' activity levels are high enough to allow sufficient space, so to speak, for moderate within-country inequalities to exist without pushing the lowest consumers below an appropriate sufficiency floor. But this certainly needs some thought and a discussion in the main text. Pauliuk (2024) deserves a mention here, as they relate sufficiency floors to average consumption levels via the Gini coefficient and a simple equation:

<https://www.sciencedirect.com/science/article/pii/S0921800924000582>

Minor comments:

A few sentence in the introduction could benefit for rewriting. The meaning of Line 4, for example, ‘...but lack the social and environmental dimension beyond GHG emissions’, is unclear to me, and I feel similarly about line 6 and 7. The sentence on lines 28-32 may also prove confusing to some, mostly as it is too long. Finally, on line 37, it is not clear if the stated 40% contribution of sufficiency is 40% of the just- mentioned 50%, or 40% absolutely (thus accounting for 80% of the 50% reduction).

The tracked changes have caused typos in a couple of place (lines 47 and 59, may be more I didn’t spot).

On line 78: ‘comparison of these trajectories’ with what, and why?

On line 105-106: ‘per capita approach’, do you mean equal per capita approach?

Line 148: remove ‘only’ or ‘just’.

Line 157: ‘almost entirely reduced’ does not make sense; perhaps change to ‘almost entirely eliminated’?

Line 181 onwards: I think framing sufficiency as entirely a policy question misses the extensive social and cultural changes that are necessary, at least for a transition to be democratic.

Lines 211-212: I think ‘for an inclusive consideration of mitigation strategies’ is redundant and could be removed.

Dear Reviewers,

Thank you very much for reviewing. We very much appreciate your very valuable work.

In the following we answer each of the comments in blue.

Reviewer #3 (Remarks to the Author):

Dear authors,

Thanks for your careful efforts to improve the manuscript. The revised version appropriately clarifies my doubts and is much more clear than the original submission. Well done!

ANSWER: Many thanks for your comprehensive review, which has greatly helped to improve the paper.

The manuscript would still benefit from proof-reading to correct the spelling mistakes (e.g. L48 “Deliverontribution”, L61 “mproved”, L362 “he calculation”),

ANSWER: Thanks a lot for spotting the spelling mistakes, we have corrected those.

unclear sentences (e.g. INTRODUCTION L36-38. “50% of final energy demand reduction is realised (compared to 2019), of which at least 40% are due to sufficiency.” 40% reduction of “final energy demand” or “final energy demand reduction”?)

ANSWER: We have gone thoroughly through the paper and have clarified the unclear sentences in the text. Regarding the 40%-reduction sentence, we have clarified it in text: “50% of final energy demand reduction is realised (compared to 2019). At least 40% of this final energy demand reduction is due to sufficiency, ...”

and language and choice of words (e.g. L161-162 “energy imports from outside of Europe are almost entirely reduced”. Replace “reduced” by some other word.).

ANSWER: We have gone thoroughly through the paper to improve the choice of words. We have replaced “reduced” with “eliminated”.

Reviewer #4 (Remarks to the Author):

The paper successfully addresses key aspects such as methodology, results, implications, and contributions to the field, making it suitable for publication.

ANSWER: Many thanks for the review and the recommendation for publication.

Reviewer #5 (Remarks to the Author):

This is in my opinion an important and timely contribution to the climate mitigation literature. It explores (thoroughly) a fair and feasible pathway for Europe to live up to its climate responsibilities and devotes the necessary attention to sufficiency alongside the more traditional approaches' focus on clean energy and efficient technologies. I would be happy to see it published.

ANSWER: Thank you very much for this assessment and for the valuable comments, which we will answer individually below and describe how we have incorporated the suggestions in the paper.

I have two major comments, which can be addressed via discussion and do not necessitate further analysis, and various other minor comments.

Major comments:

1. My first major comment involves the use of Decent Living Standards (DLS). These feature quite heavily in the paper's introduction, but it is not entirely clear to me how they enter into the authors' scenario. There are two separate issues here:

One is more minor, and it is that I am not sure how DLS relate to the bottom of the corridors for residential floor space and mobility shown on Figure 3? The values for residential floor space in the DLS literature are certainly much lower than the ~35 m²/cap shown on Figure 3a.

ANSWER:

The value proposed in the Millward-Hopkins (2020) scenario of a floor area between 15 to 20 m² (considering the diversity of household size) could be considered as a theoretical social lower bound for the corridor in a sufficiency approach. Indeed, this scenario ensures that this value could allow to live a decent life. Those values relate to an individual minimum. However, given the current values observed in Europe and the trend of other scenarios, a target for the lower bound between 32 and 37m²/person is proposed – on average in the entire population. This target is considered more realistic and consistent with:

- The decent living standard defined by Rao and Min (2018) in urban and rural areas for China.
- The observed level in European countries such as France where the value of 40m²/pers allows a sufficient standard of living even if there is still a significant over-occupation rate (slightly below 10%).
- The estimations of other European level scenarios considering 37m²/pers. in 2050 as the most ambitious target in the EUCALC and NetZero pathways.

This overestimation of the lower bound compared to Milward-Hopkins (2020) allows also for a certain flexibility in order to guarantee a decent standard of living for all, in particular to take into account some inequalities in the standard of living in the population that might persist.

Many parameters (speed of renewal of the building stock, reduction in the size of households,

decrease in the population, etc.) reduce the possibilities to rapidly reduce the floor area per person. An average decrease of 0.4%/year (CAGR) between 2015 and 2050 already seems very ambitious.

As this full explanation would be too long for the main text, we have just added the following general explanation in the section “Bottom-up and contraction and convergence approach”:
“Decent living standards define minimum living standards on an individual level, total energy consumption projected on national levels on this basis thus reflect a decent living standard-based theoretical minimum. If the distribution of service level-indicators as e.g. m²/cap includes also higher levels due to unequal distribution in reality, this will unequivocally raise national averages. We thus do not apply theoretical minima, but levels for national averages that we consider to ensure decent living standards for all while being realistic to be achieved.”

The second issue is that much of the DLS inventory refers to goods and services at the product level – phones and laptops, clothes and washing facilities – or services for which Europe relies upon substantial importation of products – i.e., healthcare. If I understand correctly, the current work instead only looks at industry with respect to production of key materials – e.g., steel, cement, paper. A proper DLS analysis, then, requires analysis of specific products, but this is very hard to do thoroughly with a large full-economy model as the authors use. I would thus suggest instead adding this important point to the discussion somewhere (perhaps briefly in the main text and in more detail in the limitations section of the methods?)

ANSWER:

Thanks for pointing this out. It is correct that the industry indicators of the CLEVER scenario are on the level of amount of steel etc. produced and are not directly calculated based on the amount of products on the service level like clothes, phones per person etc.. The amounts of material produced in the CLEVER scenario are however informed by other scenarios which consider detailed analysis of consumption of goods with a high level of disaggregation, specifically the most recent French negaWatt scenario. Thus, results from the very detailed French scenario are adapted to the context of other European countries, which is the basis for the production corridors for other countries. The historic rate of production over consumption of material is kept in the CLEVER scenario.

In the main text, subsection “Sufficiency and efficiency impact in the industry sector” we have added *“The production volumes are based on sufficiency-oriented consumption at the product and service level, but refer to other studies; they are not modelled at the product level in this study. The historic rate of production over consumption of material is kept in the CLEVER scenario.”*

In the method subsection “Development of national trajectories” we have added *“The industry indicators of the CLEVER scenario are on the level of amount of material produced and are not directly calculated based on the amount of products on the service level like clothes, phones per person etc.. The amounts of material produced in the CLEVER scenario are however informed*

by other scenarios which consider detailed analysis of consumption of goods with a high level of disaggregation, specifically the most recent French sufficiency scenario (negawatt 2022). Thus, results from detailed French scenario are adapted to the context of other European countries, which is the basis for the production corridors for other countries. The historic rate of production over consumption of material is kept in the CLEVER scenario.”

In the section “Limitations of this work” we have added “In this study, territorial-based accounting of emissions has been applied and the historic rate of production over consumption of goods and material has been kept. For a fully consistent approach referring to decent living standards also on a consumption goods level, consumption-based accounting of emissions would be required in combination. This would however imply additional model types.”

The other consequent implication that deserves mentioning is that while the scenario the authors develop may avoid European imports of energy, is it unlikely to avoid imports of manufactured goods – e.g., vehicles, home appliances and electronics, clothing, cleaning products. So there remains potential justice issues, as such imports may continue to happen through unequal exchange, if Europe retains its current purchasing power advantages over the Global south, which allow the exploitation of low-wage labour via globalised supply chains. Lines 159-161 may be a good place to highlight this.

ANSWER:

We fully agree with your point. We have added “Another aspect of a just energy transition that goes beyond the quantitative analysis of this paper, would be to consider the role of imported manufactured goods and not only the import of energy carriers as potential justice issues might continuously occur due to unequal exchange between European countries and the Global South”. at the end of the result section.

But there is a lower consumption (there are fewer products and their size is better adapted to final uses, then usually they are smaller (e.g. vehicles)). Therefore, given that the ratio of consumption over production is maintained constant, imports of manufactured goods remain but they are lower in absolute terms.

2. The second major comment I have is that there is no consideration of inequalities within countries, from what I can see. Sufficiency floors in terms of m²/cap of living space or pkm of mobility, must apply to individuals to be meaningful, and a national average exceeding a sufficiency floor does not guarantee decent living standards are available to all. Considering Figure 3, I suspect the authors’ activity levels are high enough to allow sufficient space, so to speak, for moderate within-country inequalities to exist without pushing the lowest consumers below an appropriate sufficiency floor. But this certainly needs some thought and a discussion in the main text. Pauliuk (2024) deserves a mention here, as they relate sufficiency floors to average consumption levels via the Gini coefficient and a simple equation:

<https://www.sciencedirect.com/science/article/pii/S0921800924000582>

ANSWER: Thank you very much for the very valuable reference, we have included it now in the paper. And yes, we fully agree that national averages do not reflect on inequalities within countries. This is one reason why the minimum levels of the corridors are above DLS (see also our answer to your first comment on DLS with respect to the corridor for living space per person). We have however not done any more quantitative analysis on distribution within countries. We do touch upon this topic. We however deal with the topic in the context of the policy measures that are proposed to achieve the CLEVER scenario. Some of those explicitly aim at ensuring that every individual within the country achieves the basis for a good life.

We have added the following reflection in the discussion section:

“Sufficiency is adopted within the modelling framework at international level but intra-country inequalities are not explored. By reducing national average energy consumption, energy demand reduction has the potential to lead to the lowest consumers falling below decent living standards without addressing energy inequality (Millward-Hopkins and Johnson 2023). Pauliuk (2024) presents a framework for assessing suitable levels of inequality whilst ensuring decent living standards are met by all, which can be applied to a variety of indicators, including the ones adopted in the CLEVER scenario. Although the CLEVER scenario adopts sufficiency floors above that recommended limits, to allow for sufficient space for the lowest consumers, future work should assess whether this is feasible. Future work should also assess how policy measures must be designed not only to ensure that the sufficiency corridor is achieved on average per country, but also to ensure fair distribution within countries so that minimum standards are achieved for each person.”

In addition, we mention the relationship between fair distribution within the countries and the listed policy measures in the discussion section.

“Some of the policy measures are not only aimed at achieving the sufficiency corridor per country in Europe, but also explicitly at ensuring that every individual within the country achieves the basis for a good life.”

Minor comments:

A few sentence in the introduction could benefit for rewriting. The meaning of Line 4, for example, ‘...but lack the social and environmental dimension beyond GHG emissions’, is unclear to me, and I feel similarly about line 6 and 7.

ANSWER:

We fully agree and have rewritten Line 1-9 of the introduction.

The sentence on lines 28-32 may also prove confusing to some, mostly as it is too long.

ANSWER:

We have split and rewritten the sentence.

Finally, on line 37, it is not clear if the stated 40% contribution of sufficiency is 40% of the just-mentioned 50%, or 40% absolutely (thus accounting for 80% of the 50% reduction).

ANSWER:

We have clarified this in the text: "50% of final energy demand reduction is realised (compared to 2019). At least 40% of this final energy demand reduction are due to sufficiency, ..."

The tracked changes have caused typos in a couple of places (lines 47 and 59, maybe more I didn't spot).

ANSWER: Thanks for spotting!

On line 78: 'comparison of these trajectories' with what, and why?

ANSWER: We have clarified in the text that the ambition levels of the national trajectories by one another and by other references have been compared to provide a basis for a harmonised European scenario and to check whether the ambitions are sufficient and realistic.

On line 105-106: 'per capita approach', do you mean equal per capita approach?

ANSWER: Yes, thank you.

Line 148: remove 'only' or 'just'.

ANSWER: Done.

Line 157: 'almost entirely reduced' does not make sense; perhaps change to 'almost entirely eliminated'?

ANSWER: Done.

Line 181 onwards: I think framing sufficiency as entirely a policy question misses the extensive social and cultural changes that are necessary, at least for a transition to be democratic.

ANSWER: We have concentrated on the policy aspects here as this is the factor we have analysed in this work. We think that for some of the suggested policy measures, no profound cultural or social changes are required as they resonate with current values and provide extensive co-benefits. However, we do agree to your point that some of the more profound policy measures would need to resonate with more extensive social and cultural changes. We have reflected on that in the discussion section. We have furthermore added our thoughts that the strong focus of citizens' assemblies on sufficiency measures to combat climate change can also be interpreted as an indication that the cultural and social changes required for comprehensive sufficiency have already begun.

Lines 211-212: I think 'for an inclusive consideration of mitigation strategies' is redundant and could be removed.

ANSWER: Done.

REVIEWER COMMENTS

Reviewer #5 (Remarks to the Author):

Thank you to the authors for replying to my concerns thoroughly. I am particularly glad to hear that the answers to two of my major concerns is that they essentially offset each other – that the reason activity levels are set to exceed decent living standards is to account for within-country inequalities, which I was concerned had been ignored.

My only remaining issue is that I think the answer the authors have offered in their responses here regarding within-country inequalities is not so clear in the paper itself.

Specifically, I feel the authors' additional text (page 3, bottom) is not completely clear. It would be useful in my opinion to have a summary sentence that says very simply something like 'we set national average activity-levels to exceed decent living thresholds so as to allow space for within-country inequalities'.

I would also potentially go a step further in the later discussion (page 10). Your residential floor space assumptions of 32-37m²/cap are about twice the DLS threshold of 15-20m²/cap, and from the Pauliuk paper I previously mentioned, when average activity levels are double a decent living threshold this implies a Gini of around 0.33. So if I understand correctly, your assumptions leave room for a floor space Gini of around 0.33. If this is the case, I think it would be useful context for the reader, and a similar estimate for mobility would be useful, too.

Otherwise, I'm satisfied with the authors' responses and look forward to seeing this valuable work published.

Dear Reviewer,

Thank you for your further comments, which have helped us to clarify the important issue of unequal distribution within countries in our paper.

In the following we answer each of the comments in blue.

Reviewer #5 (Remarks to the Author):

Thank you to the authors for replying to my concerns thoroughly. I am particularly glad to hear that the answers to two of my major concerns is that they essentially offset each other – that the reason activity levels are set to exceed decent living standards is to account for within-country inequalities, which I was concerned had been ignored.

ANSWER: Yes, thanks for bringing those points up, which we had missed to explain in the manuscript.

My only remaining issue is that I think the answer the authors have offered in their responses here regarding within-country inequalities is not so clear in the paper itself.

Specifically, I feel the authors' addition text (page 3, bottom) is not completely clear. It would be useful in my opinion to have a summary sentence that says very simply something like 'we set national average activity-levels to exceed decent living thresholds so as to allow space for within-country inequalities'.

ANSWER: Thank you for this suggestion. We have adapted the wording and have added a summarising sentence very similar to what you have proposed:

“Decent living standards define minimum living standards on an individual level (i.e., activity levels per capita). At the national level, total energy consumption projections based on these activity levels would reflect a theoretical minimum level of consumption for a given country. However, due to intra-country inequality of service-level-indicators (e.g. m²/cap), this would result in a proportion of the population falling below decent living minimum standards. We thus do not apply theoretical minima, but set national average activity-levels to exceed decent living thresholds to account for within-country inequalities.”

I would also potentially go a step further in the later discussion (page 10). Your residential floor space assumptions of 32-37m²/cap are about twice the DLS threshold of 15-20m²/cap, and from the Pauliuk paper I previously mentioned, when average activity levels are double a decent living threshold this implies a Gini of around 0.33. So if I understand correctly, your assumptions leave room for a floor space Gini of around 0.33. If this is the case, I think it would be useful context for the reader, and a similar estimate for mobility would be useful, too.

ANSWER: We have adapted the text accordingly and have added Gini estimations for floor space and surface transport:

“For example, assumptions for average residential floor space of approximately 42m²/capita are between 2.1-2.8 times the recommended decent living minimum standards [Millward-Hopkins et al. (2020)], allowing for a floor-space Gini of between 0.36 and 0.48, according to the framework developed by Pauliuk (2024). Similarly, for surface transport the population-weighted average for surface travel for all the countries assessed would permit a passenger-kilometre Gini coefficient of up to 0.38 in relation to decent living standards suggested by Millward-Hopkins et al. (2020). Future work should build upon this by assessing how variations in income inequality impact access to energy services and the resulting compatibility with decent living energy thresholds. Sufficient analysis also needs to be given to how policy measures must be designed not only to ensure that the sufficiency corridor is achieved on average per country, but also to ensure fair distribution within countries so that minimum standards are achieved for each person.”

Otherwise, I'm satisfied with the authors' responses and look forward to seeing this valuable work published.

ANSWER: Thank you very much, that's great to hear. We appreciate your valuable review work, which further improves the paper. As a result, the important point of inequalities within countries is now hopefully more clear in the paper.